# Mapping global floods with 10 years of satellite radar data

Amit Misra [1] ✉, Kevin White[1], Simone Fobi Nsutezo[1], William Straka III [2] & Juan Lavista [1]

Floods cause extensive global damage annually, making effective monitoring essential. While satellite observations have proven invaluable for flood detection and tracking, comprehensive global flood datasets spanning extended time periods remain scarce. In this study, we introduce a deep learning flood detection model that leverages the cloud-penetrating capabilities of Sentinel-1 Synthetic Aperture Radar (SAR) satellite imagery, enabling consistent flood extent mapping through cloud cover and in both day and night conditions. By applying this model to 10 years of SAR data, we create a unique, longitudinal global flood extent dataset with predictions unaffected by cloud coverage, offering comprehensive and consistent insights into historically flood-prone areas over the past decade. We use our model predictions to identify historically flood-prone areas in Ethiopia and demonstrate real-time disaster response capabilities during the May 2024 floods in Kenya. Additionally, our longitudinal analysis reveals potential increasing trends in global flood extent over time, although further validation is required to explore links to climate change. To maximize impact, we provide public access to both our model predictions and a code repository, empowering researchers and practitioners worldwide to advance flood monitoring and enhance disaster response strategies.

Floods are the deadliest natural hazards, striking numerous regions in the world each year[1]. Floods cause 40 billion dollars (2015 USD) in damages annually[2] and affected 2.5 billion people between 1994 and 2014[3]. Furthermore, the population of people living in flood-prone areas is increasing due to migration and population growth[4,5]. All of these impacts are expected to become even more severe with climate change[6].

A key factor in understanding and mitigating impacts from flooding is knowing where flooding occurs on a regular basis. Accurate flood extent mapping provides essential data for various purposes. It helps urban planners design resilient infrastructure, aids in developing early warning systems, and supports insurance companies and policymakers in assessing risks and allocating resources. By understanding past flooding events, communities can better prepare for future occurrences, leading to safer and more resilient living environments. While mapping flood extent is important, doing this via on-the-ground efforts is often challenging, especially in developing countries, where resources for this time-intensive work are scarce. In addition, ground-based assessments are often for small areas.

Satellite data offers a powerful solution for mapping flood extent at scale. Two primary types of sensors are commonly used for this purpose: optical/infrared and Synthetic Aperture Radar (SAR). Optical/infrared sensors passively capture reflected light, yielding familiar photographic images with benefits like wide availability and frequent observations at various resolutions. However, their effectiveness is limited by cloud cover and a dependence on daylight. In contrast, SAR actively emits microwave signals and records their reflections, offering advantages such as being able to penetrate through cloud cover and

[1]Microsoft AI for Good Research Lab, Redmond, WA, USA. [2]Cooperative Institute for Meteorological Satellite Studies, University of Wisconsin-Madison, Madison, WI, USA. ✉e-mail: amitmisra@microsoft.com

operate in both day and night conditions. While SAR is often described as all-weather, its signal may be affected during extremely heavy rainfall[7,8]. However, SAR satellites like Sentinel-1 typically provide observations every 6–12 days for a given location, whereas optical satellite constellations have revisit times ranging from daily to several days, often enabling more frequent coverage than SAR. This lower temporal frequency can affect our ability to capture the full dynamics of flood events with SAR, particularly flash floods and peak flood extent that may occur between SAR observations. Additionally, both technologies face challenges in complex terrain such as narrow valleys, which are common worldwide.

To understand where flooding occurs regularly, researchers have attempted to track flood events systematically over time. Previous work on tracking global flood events over time has primarily relied on optical and infrared imagery, despite limitations from cloud coverage. One notable study[4] used relatively coarse resolution (250 m) visible and infrared data from MODIS (Moderate Resolution Imaging Spectroradiometer) to map known flood events from the Dartmouth Flood Observatory[9], a curated catalog of flood events. Similarly, the Global Surface Water (GSW) maps uses Landsat data (30 m resolution optical and infrared imagery) to track surface water and its changes over time[10], though this study was not specifically focused on flooding. Neither the MODIS- or Landsat-based archives are being updated over time. In addition to these global, temporal datasets, there are existing tools for real-time flood mapping using optical and infrared imagery[11–13]. However, all of these approaches face a fundamental challenge in that cloud coverage often obscures flood events.

In light of the challenge of cloud coverage, SAR imagery offers a significant advantage for flood detection because SAR microwave signals penetrate cloud cover and can be used in both day and night conditions. SAR's effectiveness in flood mapping stems from the distinct backscatter signature of water surfaces, which appear dark in SAR imagery due to specular reflection of the radar signal away from the sensor. This characteristic makes SAR particularly suitable for distinguishing water from other land cover types. Researchers have successfully applied various techniques to Sentinel-1 imagery for flood mapping, including threshold-based approaches, machine learning methods, and more recently, deep learning algorithms[14]. Combined with its high resolution capabilities (as fine as 10 m for Sentinel-1), SAR has become a valuable tool for flood mapping. SAR's effectiveness in detecting flooding and its cloud-penetrating capabilities were highlighted in a comparison of flood detection between Sentinel-1 (SAR) and Sentinel-2 (optical) satellites over Europe, with SAR imagery detecting 58% of flood events while optical imagery captured only 28%, given the same number of satellites[15].

Given these capabilities, SAR data has been used extensively for flood mapping[16–18], especially over Europe and Canada, where satellite ownership enables more frequent observations than other regions in the world. However, unlike the comprehensive, multi-year datasets created using MODIS and Landsat imagery, there has not been an effort to create a similar dataset of flood extent using SAR imagery. While the Copernicus Global Flood Monitoring (GFM) system provides valuable SAR-based flood maps[19], it is primarily designed for analyzing individual flood events rather than producing aggregate maps showing all flood detections across multiple years or tracking longitudinal flooding trends. Therefore, there remains a need for an analysis of flooding patterns over extended periods. A dataset that aggregates global SAR flood detections over multiple years, while carefully accounting for challenges like false positives, could provide additional insights into the spatial and temporal patterns of flooding.

Our aim is to address this gap by building a neural network model to detect flood extent from Sentinel-1 SAR imagery and apply this model to 10 years of available SAR data to provide a global flood extent database over time. Our model uses a change detection approach, comparing pairs of SAR images acquired before and during potential flood events to identify inundated areas. We focus solely on SAR to ensure consistent detection through cloud cover and in both day and night conditions, which is beneficial for creating reliable aggregate flood maps and enabling unbiased temporal analysis. We account for false positives with auxiliary datasets such as soil moisture, digital elevation models, temperature and land cover mappings.

Applying the model to a decade of Sentinel-1 SAR data enables the creation of a unique global dataset that supports several important use cases:

1.  A comprehensive historical baseline: We generate a high-resolution map identifying historically flood-prone areas globally over the past decade. Leveraging SAR's ability to observe through clouds, this dataset provides a consistent perspective that complements optical datasets often hindered by cloud cover during flood events. While historical flooding is not a perfect predictor of future risk, particularly as climate change and other dynamic factors (e.g., land use change) may alter future patterns, this baseline can inform risk assessments, mitigation planning, and resilient infrastructure development.

2.  Enhanced rapid response: The underlying model and processing pipeline can provide near-real-time flood extent maps during crises, serving as a valuable tool for disaster response teams.

3.  Observation-based trend analysis: The longitudinal nature of our datasets enables analysis of potential trends in flood extent over time. Although the 10-year span limits definitive climate attribution, this SAR-based approach offers a globally consistent, observation-driven view of flood dynamics over time. It also helps mitigate biases associated with report-based trend studies[20,21] and provides a foundation for ongoing monitoring as the satellite record expands.

## Results

### Global flood map and application to Ethiopia

We aggregated all flood detections over the 10 years of Sentinel 1 SAR data available at the time of our analysis (Oct 2014–Sep 2024) to create a global flood extent map, as shown in Fig. 1. While we are able to track flooding detected on specific dates and map flood rates over time, here we present flood extent as a binary output for ease of visualization, marking only whether or not flooding has been detected in that pixel after removing potential false positives. In SAR imagery, there are multiple potential causes of false positives that need to be accounted for when running a model over an extended time period and not just for specific flood events (see "Methods" Section for more details). Additionally, while we run our model at a 20 m spatial resolution, here we create the map at 250 m resolution to aid in visualization.

After creating the global flood map, we developed an exclusion mask to identify areas where flood detection may be unreliable or prone to false positives. This mask covers urban areas, where building interference makes flood detection challenging, arid regions where certain surface features can create false positives, and areas with rough terrain. In our visualizations, these excluded areas are shown in gray. This masking approach, similar to that used by the Copernicus GFM system, helps users understand where our flood detection capabilities may be limited. More details on the exclusion mask logic can be found in the "Methods" section.

To contextualize our flood detection approach, we compared our results with two existing GSW datasets: the Landsat-based GSW dataset[10] and a MODIS-based dataset[4] (see Table 1). For the GSW dataset, we considered anything with water occurrence less than 50% as a flood-prone area. Our analysis shows significant increases in detected flood extent compared to these existing datasets. Globally, we estimate that our results increase areas with detected historical flooding by 71%. Given that our dataset's timespan (2014–2024) is largely covered in the GSW (1984–2021) and MODIS (2000–2018)

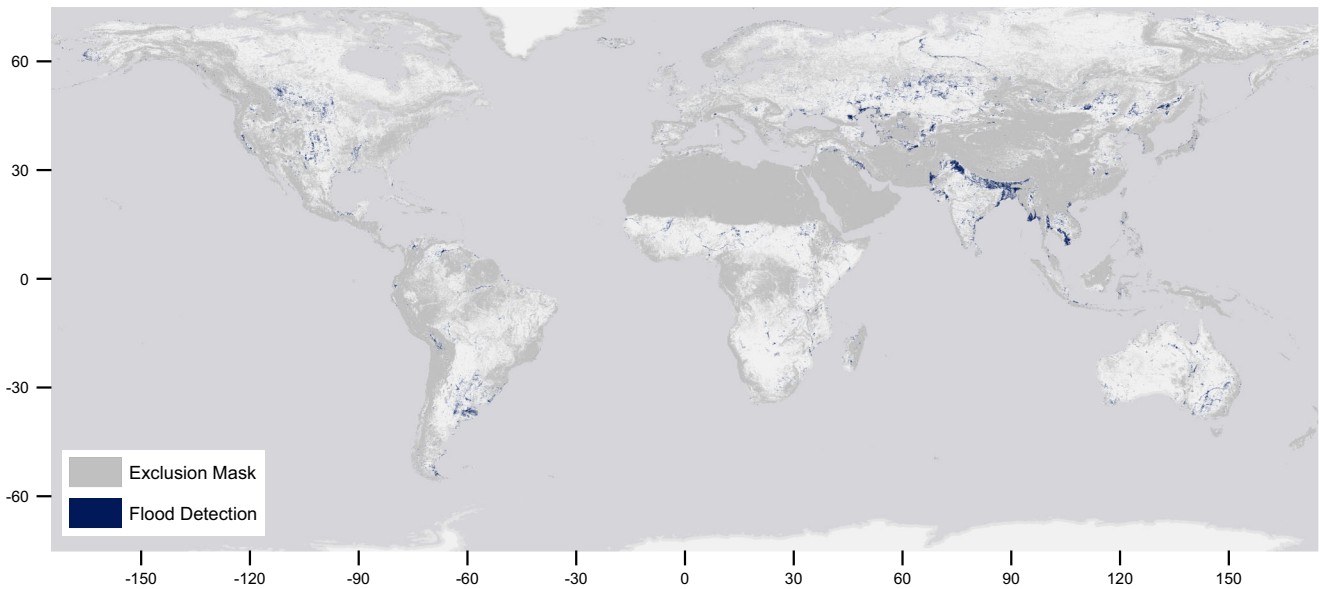

**Fig. 1 | Global flood map.** Aggregated global flood extent map as detected by our deep learning model applied to 10 years of Sentinel-1 SAR data (October 2014–Sep 2024). Blue areas indicate locations where flooding was detected at least once during this period, shown at 250-m resolution. Darker gray areas represent the exclusion mask, indicating regions where flood detection may be unreliable due to urban development, steep terrain, or arid conditions. Areas without color showed no flooding during the observation period. This map highlights historically flood-prone regions identified by cloud-penetrating SAR data.

## Table 1 | Comparison of our flood detection method with existing datasets

| Region | Additional Flood Area Identified | Overlap With Existing Flood Detections | |
|---|---|---|---|
| | | **GSW** | **MODIS** |
| Global | +71% | 35% (48%') | 28% (33%') |
| Africa | +90% | 44% (57%') | 33% (37%') |
| Ethiopia | +194% | 51% (78%') | 21% (44%') |
| Semera | +96% | 59% (68%') | 31% (41%') |
| Dolo Ado | +1013% | 59% (78%') | 72% (76%') |

Our model successfully detects a substantial portion of flood-prone areas identified in existing datasets while revealing significant additional flood extent. The first column shows the percentage of additional flood-prone area identified by our model relative to a combination of GSW and MODIS flood detections. Our results identify considerably more flood extent than previous datasets, particularly in Ethiopia and its sub-regions. The next two columns show the percentage of flood areas from established datasets (GSW and MODIS) that our model detected. Numbers in parentheses (') indicate detection rates when excluding areas where SAR detection may be unreliable (e.g., urban areas, steep terrain).For context, when comparing the existing datasets to each other, GSW captures 36% of MODIS-identified flood areas, while MODIS captures 23% of GSW-identified areas - similar to the overlap rates we observe with our model.
*GSW* Global Surface Water dataset, *MODIS* Moderate Resolution Imaging Spectroradiometer dataset.

datasets, this 71% increase suggests our approach is not merely capturing recent events, but rather detecting flood-prone areas that optical sensors missed during the same observation periods.

We also find strong overlap in locations where previous methods have detected flooding. When examining areas where the GSW dataset identifies flooding, our method detects flooding in 35% of these locations, increasing to 48% when restricting to areas outside our exclusion mask. The comparison with MODIS-based maps shows similar patterns, with our method detecting flooding in 28% of MODIS-identified flood areas (33% outside the exclusion mask). This is similar to the overlap between MODIS and GSW compared to each other (36% and 20%). Note that we do not expect perfect overlap between any of these datasets due to inherent differences in observation time spans, temporal resolution, and sensing modalities (optical/infrared vs SAR).

While we can create this global map, the primary significance is being able to go deeper and analyze any location on the globe using the same, scalable methodology. To illustrate this capability, we examined flood patterns in Ethiopia, a country that reflects broader trends observed across Africa. Across the continent, our model detects a 90% increase in flood extent compared to existing datasets. We focused on Ethiopia because we were able to work with organizations within the country with deep domain knowledge about expected flood patterns and get qualitative validation of the insights shown here[22]. At a country scale, our flood map identifies both well-known flood areas - such as regions near the Awash and Shabelle rivers and around Lake Tana in the northwest - and reveals additional flood-prone regions not captured in existing datasets, as shown in Fig. 2. We estimate that our results increase the flood detections in Ethiopia by 194%—nearly a 3× increase over existing methods.

In Fig. 3, we highlight two areas where our model reveals additional flood-prone regions not captured in existing MODIS and Landsat-based datasets: Semera in the Awash River Basin and Dolo Ado along the Ganale River. Near Semera, we see a 96% increase in flood-prone areas. Several factors contribute to our confidence that this represents improved flood detection rather than noise. First, SAR's ability to penetrate cloud cover enables observation during flood events often missed by optical sensors, providing a more temporally complete view. Second, the model's robust performance—particularly its high recall validated on the geographically diverse Kuro Siwo benchmark (see Methods subsection on "Model Validation") —supports its ability to detect true flooding when present. Third, qualitative validation from local experts in Ethiopia with deep knowledge of regional flood behavior provides corroborating support that many of these detections are consistent with known flood-prone areas[22]. While our model identifies substantially more extent, we also note regions—especially wetlands near lakes—where optical datasets detect water not captured by our model, underscoring the complementarity of different sensing modalities.

In Dolo Ado, the increase in detected flood extent is even more pronounced, exceeding 1000% relative to the combined extent from MODIS and GSW. While the large flood event in November 2023, which falls outside the coverage of the comparison datasets, contributes significantly to this increase, our model also captured extensive

flooding prior to 2021 that is not observed in the optical datasets. These additional detections, supported by the same validation factors described above, suggest our model identifies genuine flood events historically underrepresented in global optical datasets.

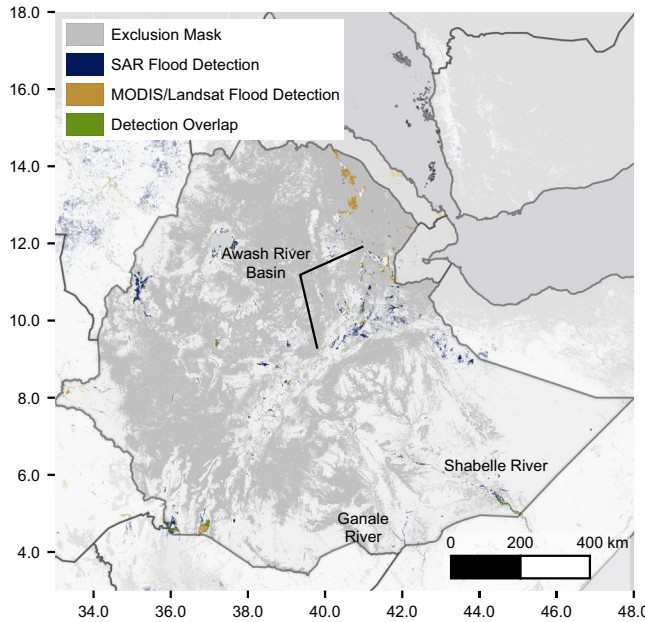

Fig. 2 | **Comparison of flood extent maps for Ethiopia.** Comparison of flood extent mappings over Ethiopia from multiple satellite sources. Blue areas show SAR flood detections from our deep-learning model (2014–2024), while orange areas represent historical flood extent from MODIS and Landsat optical and near-infrared imagery (1984–2021). Green areas indicate agreement between SAR and optical datasets. The map reveals both consistencies and complementarity between detection methods, with notable flood-prone areas identified along the Shabelle, Ganale, and Awash rivers. Darker gray areas indicate regions where SAR flood detection may be unreliable due to terrain or land cover characteristics.

This enhanced flood detection capability significantly influences our understanding of flood risks to critical areas, such as cropland. By overlaying our flood map with land use/land cover data from ESRI (Environmental Systems Research Institute)[23], we assessed the extent of cropland at risk near Semera, as shown in Fig. 4a. We estimate that 19% of cropland in this region falls within historically flooded areas according to our map, compared to 7% in the GSW dataset and 2% in the MODIS-based dataset. The contrast is even more pronounced in Dolo Ado, where our model identifies 52% of cropland in flood-prone areas versus just 1–3% in existing datasets. These regions primarily rely on rainfed agriculture for staple crops like sorghum and maize, making unplanned flooding a significant risk for the local population who depend on subsistence farming.

With the high resolution flood map, we can zoom in further to identify areas with historical flooding at even finer granularity, as in Fig. 4b, which provides a more detailed view of one particular area of Semera. Building on the previously stated assumption that historical flood patterns can inform future risk assessments, we can use these detailed maps to identify specific areas that may be vulnerable to future flooding. Given the importance of local agriculture for the populations in this area, it is important to understand which areas could be at risk of future flooding so that government agencies and other stakeholders, such as non-governmental aid organizations, would be able to target mitigation efforts as well as targeted infrastructure improvements or policies on future settlement and agriculture.

While the example above was for a specific region in Ethiopia, the methodology employed could be used anywhere in the world. This capability to produce high-resolution, detailed flood risk assessments anywhere in the world underscores the potential of our approach to significantly enhance flood preparedness and mitigation efforts across diverse geographic and socio-economic contexts. By enabling precise identification of flood-prone areas, our maps can support targeted interventions, ultimately contributing to more resilient communities worldwide.

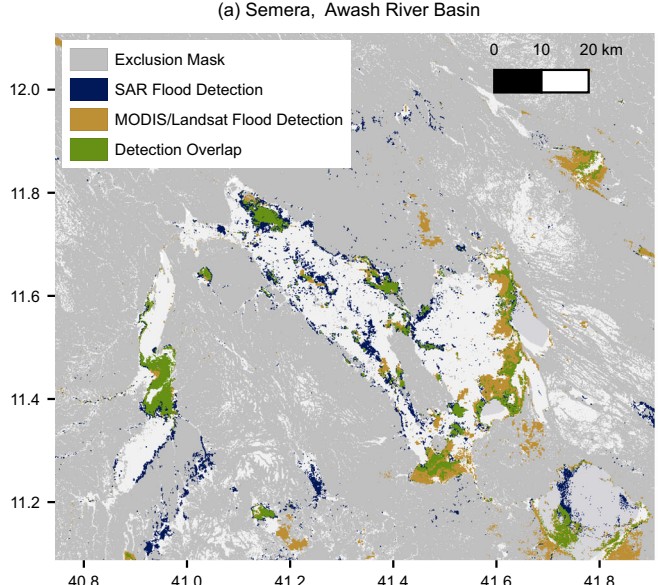

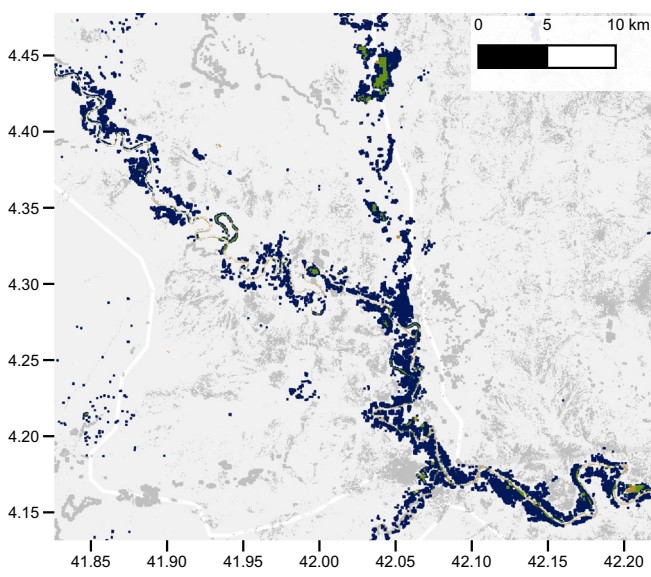

Fig. 3 | **Detailed flood detection comparison in key Ethiopian regions.** Comparison of flood extent mappings in two flood-prone regions of Ethiopia: **a** Semera in the Awash River Basin and **b** Dolo Ado along the Ganale River. Blue areas show Sentinel-1 SAR flood detections from our deep-learning model (2014–2024), orange areas represent MODIS/Landsat flood extent (1984–2021), and green indicates

agreement between datasets. Both regions demonstrate increased flood detection capabilities from our method, with significant amounts of flooding detected only by our model (blue). Darker gray areas indicate regions where flood detection may be unreliable.

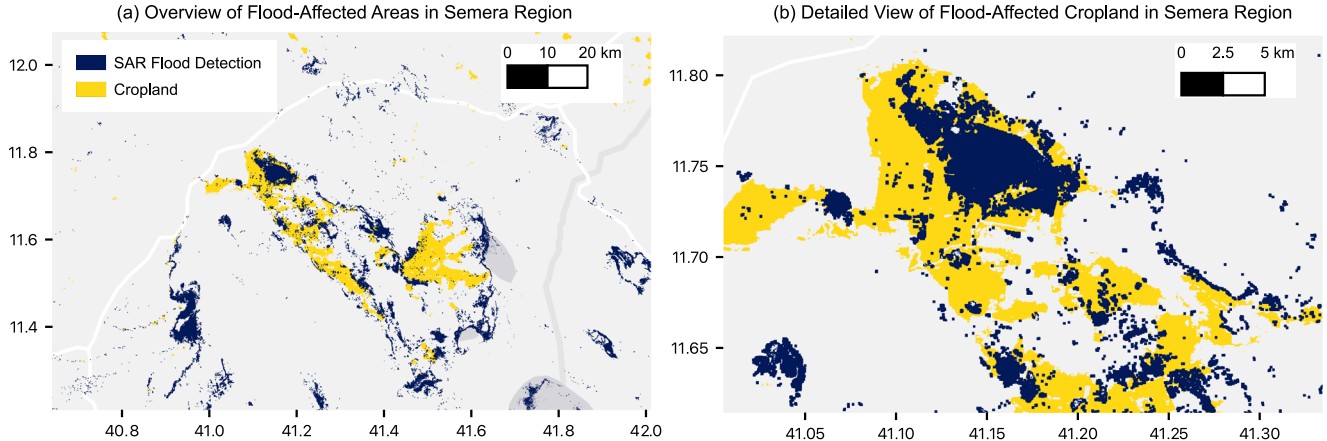

Fig. 4 | **Semera flood map. a** Overlay of cropland and flood extent maps near Semera, Ethiopia. Analysis reveals that ~19% of cropland near Semera is within historically flood-affected zones according to our flood map. **b** Detailed view of a specific area in the northwest part of the overview, illustrating the overlap between cropland (yellow) and flood zones (blue). The high-resolution flood map allows for the identification of specific fields at risk, demonstrating the utility of this mapping approach for agricultural planning.

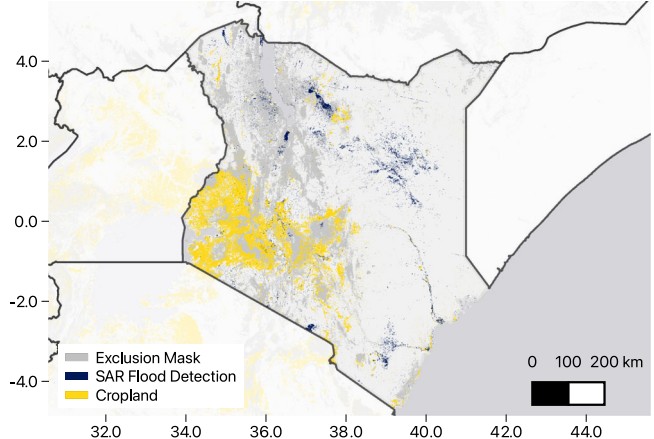

Fig. 5 | **Kenya flood map, spring 2024.** Composite flood extent map of Kenya during the 2024 floods, overlaid with cropland data. The map highlights flood-affected areas, and we estimate that ~75,000 hectares of cropland were impacted. This estimate aligns closely with official government statistics, which reported 68,000 hectares affected. The map showcases the utility of SAR data for real-time disaster response and assessment.

### Case Study: Kenya 2024 flooding and disaster response

Another application of the flood model is to be able to analyze SAR imagery for disasters. The spring rains of 2024 resulted in some of the worst flooding in Kenya's history. During the flooding, we were able to collaborate with agencies within Kenya to provide near-real-time updates of flood extent with minimal human intervention.

Figure 5 shows the flood extent map (blue) and cropland map from ESRI (yellow) over the course of the flooding between March and May 2024 for the entire country. While we updated the map daily during the flood event, the version shown is a composite of all areas where flooding was detected during this time period. As with the Ethiopia example above, we were able to overlay this with cropland maps to estimate the impact of the flooding. We estimated that roughly 75,000 hectares in the country was in or very near flooded areas (2% of all cropland in the ESRI land cover mapping for Kenya). This is roughly in line with the public government numbers of 168,000 acres/68,000 hectares affected[24].

Being able to track flooding like this in real-time is a valuable asset during a disaster. While there are existing models and tools that can do this with SAR and other satellite data, having an additional model that can be run with minimal human intervention is another source of information that can be leveraged.

### Temporal analysis of flooding trends

One major contribution from our work is the ability to track flood extent longitudinally. Climate change is expected to exacerbate flooding over time, but direct measurement of this trend is challenging. Because Sentinel-1 satellites have consistent return periods (with minor exceptions) and are minimally affected by cloud cover, we get as unbiased a view as possible of flood extent over time.

We are able to see statistical support for an increase in flooding over time, though we acknowledge the limitation of having only 10 years of data due to the operational period of the Sentinel-1 constellation. With limited data, it is difficult to attribute this to climate change, and we encourage others to extend this work and to refresh this over time as new data becomes available.

We aggregated global flood extent detections by month over the entire 10 year period. While the data is initially noisy, after removing the seasonal trend we see an increase over time. Figure 6 shows a seasonal decomposition of the overall trend. This decomposition visualizes how the flood extent signal can be separated into trend, seasonal, and residual components, helping illustrate the underlying patterns in our data. For our statistical analysis, we used a linear regression model with monthly dummy variables to control for seasonality while estimating the temporal trend. From the trend, we identified two potential data challenges: the potential outlier year of 2022, and the time before June 2017. Prior to June 2017, many of the observations were made with only one polarization channel instead of two, resulting in different rates of flood detections that we correct for before measuring the trend. The large observed flood extent in 2022 is likely due to specific flood events such as the flooding in Pakistan from June to October of that year. Given that these two factors could skew any potential estimate of the trend over time, we estimated the trend under different scenarios (see "Methods" Section for more details).

Table 2 shows the results for the different scenarios with one standard deviation. In general, we see a positive trend of a few percent per year, though in the most pessimistic case the result is not statistically significant (the result is within <2 standard deviations of 0), because of both the lower effect size and the result of removing nearly 40% of the data, which increases the uncertainty estimate. We view the middle row, where we exclude 2022 but include the earlier data and

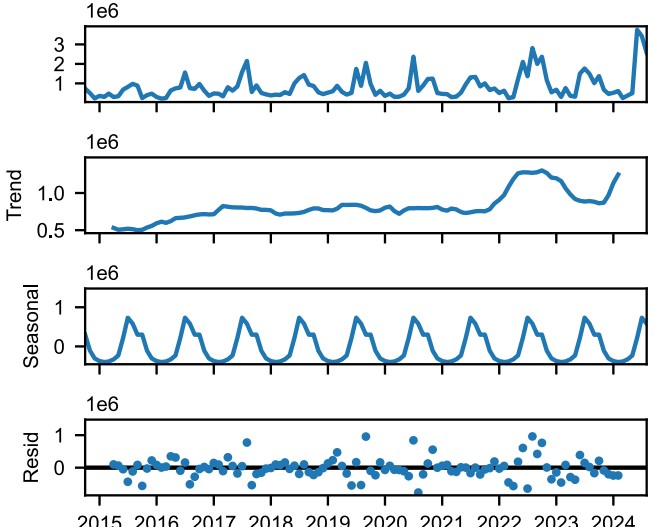

**Fig. 6 | Seasonal decomposition of flood extent trends over a decade.** The y-axis represents flooded area (in hectares) per observation. The top panel shows the raw signal from the model after removing false positives. The subsequent panels decompose this signal into trend, seasonal, and residual components. The seasonal component indicates higher flooding during northern summer months. The trend component, after removing seasonal effects, suggests an increase in flooding over time. We note that this plot is for visualization purposes only, as we fit a linear model to the data when we estimate the trend, rather than simply looking at the trend component in this plot. However, this visualization aids in understanding the underlying patterns and trends in flood extent. Source data are provided as a Source Data file.

**Table 2 | Flooding trends over time**

| Scenario | Est. Trend | *p*-val |
|---|---|---|
| All Data | 6% ± 2% | 0.0005 |
| 2022 Removed | 5% ± 2% | 0.01 |
| 2022 and pre-June 2017 Removed | 2% ± 3% | 0.5 |

Estimated trends, uncertainties (one standard deviation), and *p* values for flood extent. In general we see evidence for a positive trend, though the results are not statistically significant in the third, most pessimistic scenario. We estimated the trend under different scenarios. The first row (All Data) is where we include all time series data. In the second row, we remove 2022 as a potential outlier because it has much higher average flood extent than other years. Removing it removes the estimated trend slightly. The last row is if we exclude both 2022 and data prior to June 2017. Before
June 2017 many of the observations were made with only one of two observations channels, resulting in different rates of flood detection. This further reduces the estimated trend in flood extent over time. We view the middle row, where we only remove 2022 as a potential outlier, as our current best estimate.

see a 5% yearly increase, as our current best estimate of the change in flooding over time. This 5% increase would result in a roughly 60% increase every decade if the growth compounded, resulting in great potential loss in human life and property if not mitigated.

In addition to understanding global trends in flooding, it can be beneficial to understand where flooding is increasing or decreasing across the globe. Figure 7 shows the estimated trend for 3° by 3° tiles. Tiles that have a *p* value for the trend greater than 0.2 or have low absolute observed trends are removed to reduce noise in the plot. While there is considerable uncertainty because of the limited time range of our analysis, there are some interesting regional insights. For example, there is an area between Nigeria and Ethiopia with an estimated increase in flooding. This is correlated with predictions of increased precipitation in this region under the CMIP6 (Coupled Model Intercomparison Project Phase 6) climate scenarios, a comprehensive climate modeling framework[25]. However, other regions do not necessarily have similar increases in predicted precipitation, so any

such correlations should be interpreted with caution. These insights could be further corroborated by repeating the analysis as more data is made available and trying to combine with other data sources, such as the NOAA (National Oceanic and Atmospheric Administration) LEO Flood archive, which runs from 2012 to 2024.

## Discussion

In this paper, we presented a comprehensive approach for global flood extent mapping using SAR imagery from Sentinel-1 satellites. We have developed a method for identifying flood-affected areas by leveraging a decade of SAR data and a deep learning change detection model. Additionally, our method incorporates post-processing steps to mitigate false positives, which are often a challenge with SAR data.

Our results demonstrate the utility of our approach in creating detailed flood extent maps, which can be instrumental for identifying areas at risk of flooding and for disaster response. The case studies in Ethiopia and Kenya illustrate the practical applications of our model, from assessing flood risks to cropland to providing near-real-time updates during disaster events. Furthermore, we show that these types of global flood datasets derived from satellite data can be useful in measuring the trends in global flooding over time. We note that while our results suggest the possibility of an upward trend in flooding, more work would need to be done to confirm this result and ultimately tie it causally to climate change. The addition of optical flood products may be a benefit in helping with this analysis.

Our global flood extent map shows significant benefits over existing datasets, likely due to SAR's ability to penetrate cloud cover that often accompanies flooding events. This capability enables more complete flood detection compared to optical and infrared sensors, which can be obscured by cloud-cover during critical flooding periods. This consistent, cloud-independent observation capability enables the creation of a historical baseline of flood-prone areas, complementing optical records. For example, in Semera, our model identified substantially more flood-affected cropland than seen in the MODIS-based flood database and the Landsat-based GSW dataset, even during time periods where multiple datasets overlap. The benefits of our approach are further demonstrated near Dolo Odo in southeast Ethiopia, where we detected significant flooding along the Ganale river - flooding that was almost entirely missed in both MODIS and Landsat-based datasets. Additionally, our dataset extends through 2024, allowing us to capture recent major flood events outside the temporal coverage of existing datasets, though the improved detection rates for these regions persist even in periods where the datasets overlap.

One major contribution from our work is the release of a code repository allowing anyone to run our model. Additionally, we have released all the model predictions for every SAR image from Sentinel-1 up through September 2024, providing a valuable resource for further research and practical applications.

While these results demonstrate significant progress in flood mapping, several important limitations must be considered. Many limitations are primarily due to the inherent challenges of SAR data. Our model, while effective in many scenarios, faces difficulties in accurately detecting flash floods and urban flooding. Flash floods are challenging because of the short duration. Unless there is an observation taken during the time of the flood, our model will not capture any flooding, posing a challenge for events that often last hours or less. Urban flooding is another challenge for most satellite imagery outside of very high resolution data (sub 10 m), but especially for SAR because of the interference from buildings that makes detection of the surface difficult.

Our model also will have difficulties in correctly identifying all false positives. While we explored ways to filter out false positives for our results, we recognize that this is an open problem, especially without sufficient data from non-flood events. As discussed in the "Methods" section, arid regions, areas with rough terrain, and freeze/thaw cycles

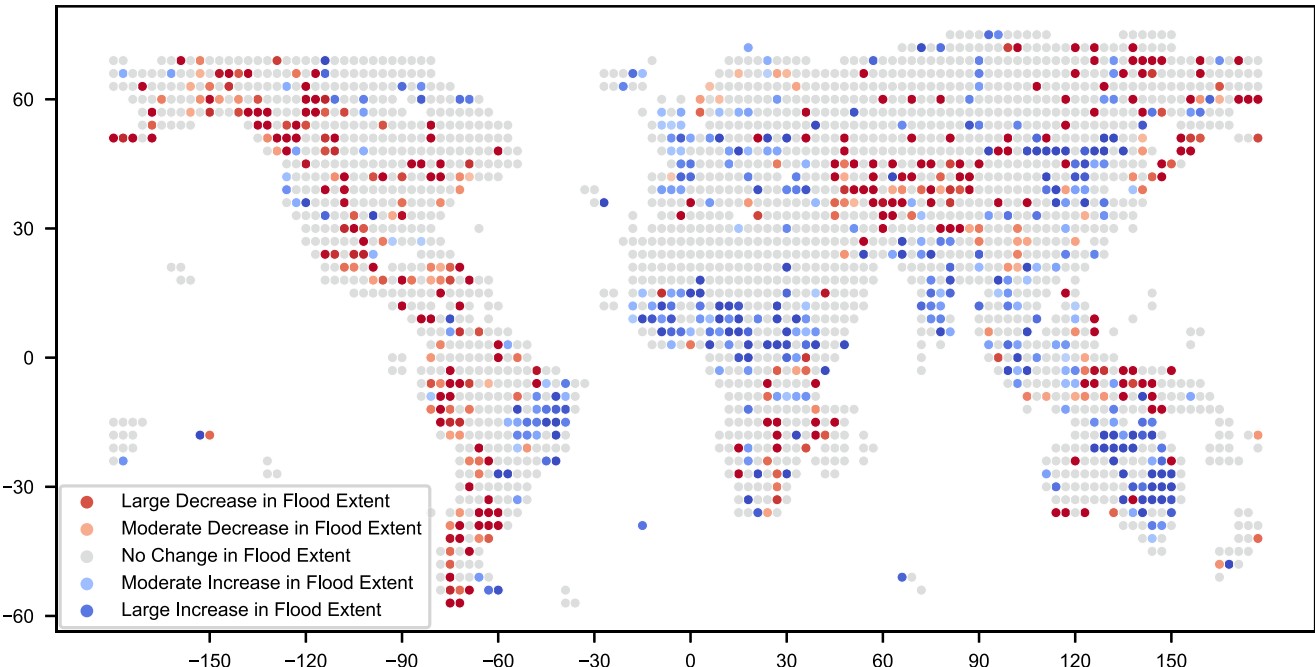

**Fig. 7 | Flood trends by region.** Global map showing regions with increasing (blue) and decreasing (red) flooding trends from 2014–2024. "Large" changes represent areas where flood extent showed net monthly increases exceeding 2% of the total land area, while "moderate" changes represent monthly increases of 1–2%. Notable increases occurred in eastern Australia and from Nigeria to Ethiopia, often characterized by major flood events rather than gradual changes. This highlights regions experiencing significant changes in flood risk, though these changes frequently reflect the impact of extreme events rather than continuous trends. Given the 10-year dataset, further research is needed to distinguish between episodic events and emerging patterns. These regional patterns merit follow-up in future work, especially to assess whether any align with climate-related drivers as longer observational records become available. Source data are provided as a Source Data file.

can create false positives. We found that adding in additional data sources in our post-processing like soil moisture, elevation and temperature greatly helps in removing false positives. However, some false positives will inevitably persist, along with true positives that may be incorrectly filtered out by our methods.

To address model limitations for end-users, we developed an exclusion mask to identify areas where our model's performance may be compromised. This mask incorporates several key factors that can impact SAR flood detection reliability. Specifically, we flag areas with steep terrain, urban development, and arid regions. Although our terrain slope measurements help identify many forested areas, explicit incorporation of forest cover data could further improve the accuracy of the exclusion mask. The mask serves as a crucial tool for end users, allowing them to better interpret our model's outputs and understand where additional verification may be needed.

While this exclusion mask helps users understand model limitations in challenging terrains, it cannot address all interpretation challenges. In particular, certain land use patterns require careful consideration when interpreting model results. A prime example is rice cultivation, where paddies are deliberately flooded as part of the agricultural cycle. While our model correctly identifies these areas as flooded, these detections represent intentional agricultural practices rather than natural flood events. Users should exercise particular caution when applying this model in regions with extensive rice cultivation or similar agricultural practices, where distinguishing between intentional and unintended flooding is crucial for proper interpretation.

These interpretation challenges highlight the importance of robust validation strategies that account for both geographic diversity and local complexities. Our current evaluation is intentionally broad, leveraging the Kuro Siwo benchmark, which spans 43 flood events across six continents. This breadth is critical for assessing performance in globally diverse conditions and for building confidence in our model's generalizability. However, recent work underscores the complementary value of event-specific, deep-dive analyses. For example, Roth et al. conducted a systematic validation of one of the algorithms used in the Copernicus GFM service across 18 flood events, with a detailed analysis of 8 of these events[26]. Their study highlighted key sensitivities for that particular radar-based flood detection algorithm, including the impact of polarization differences (VV vs. VH), vegetation cover, wind conditions, and the use of post-processing filters. Although such fine-grained evaluations fall outside the scope of our global-scale work, they offer a valuable blueprint for future efforts to diagnose and refine performance in more challenging or observationally challenging contexts.

Based on these limitations and considerations, we have identified several potential improvements to the model. One approach is to create a richer and more diverse labeled training data set. In an ideal scenario, none of the false positive filtering would be done in post processing. Instead, all of this data would be included in model training. However, this requires a substantially larger dataset, along with examples of flooding and non-flooding examples for each of the causes of false positives. For example, we would need both true positive and false positive examples in arid regions so the model could differentiate between the two. The data requirements are particularly demanding since some of the post-processing datasets are at much coarser resolution than the SAR imagery. As an example, soil moisture data is available at a resolution of 10 km. Training a model that could make best use of soil moisture data would require much more training data than what we have used here.

Another potential improvement would be to include SAR phase data to address urban flooding. SAR data has an amplitude component, which is what we use for our model, and a phase component. SAR amplitude data has inherent difficulties with capturing urban flooding

that are unlikely to be solvable with improved training data. However, there has been work on detecting urban flooding using SAR phase information[27,28], and adding it in could be a valuable addition to future work.

Other satellite imagery would also be valuable for modeling flooding. For example, while beyond the scope of the current work, Sentinel 2 and Landsat imagery have been used extensively to model flooding. Landsat data in particular is intriguing due to the longer temporal coverage, with over 5 decades of data. Fusion models, where data from different sources is combined in the flood detection algorithm itself, represent another promising direction. Fusion methods have been shown to improve flood detection using both single images[29] and change detection approaches with pre- and post-flood imagery[30]. Additionally, time series approaches for whole image-based flood detection rather than pixel-based flood detection demonstrate substantial improvements in accuracy when combining optical and SAR time series data[31]. There is also work on combining microwave satellite data with optical Landsat data[32]. While implementing such fusion approaches for long-term temporal analysis presents challenges due to cloud cover, combining SAR data with other modalities could increase flood detection accuracy in future work, particularly for individual flood events where consistent temporal coverage is less critical.

A key contribution of this work is the analysis of flooding trends over time, leveraging Sentinel-1 SAR's cloud-penetrating abilities and consistent revisit cycle. This provides a globally coherent, observation-based perspective on potential decadal changes derived purely from SAR data. However, we acknowledge that our data is over too short a timespan to draw any definitive conclusions between the observed trends and climate change. While we observed an upward trend, this could be influenced by other factors, such as natural climatic oscillations like El Niño and La Niña events, which can significantly impact weather patterns. Still, our results suggest that satellite data can play a crucial role in understanding trends in flooding over time and by geographic area. This dataset and methodology establish an essential baseline for future monitoring and offer a less biased approach compared to traditional report-based methods. As the Sentinel-1 archive grows, repeating and extending this analysis, potentially integrating other satellite data sources and hydrological information will be crucial for building a more robust understanding of long-term flooding trends and their drivers.

## Methods

Our approach for flood detection consists of two broad steps: first, running a neural network model trained on SAR data to detect flood candidates, and second, removing potential false positives using auxiliary datasets. After filtering out false positives, we used the aggregate results to create flood extent maps and to look at flooding trends over time.

### Neural network model

We used a MobileNet early fusion change detection model, which combines spatial and temporal information early in the processing pipeline to analyze SAR images. We chose MobileNet as our model architecture because it can be effectively adapted for pixel-level classification tasks in satellite imagery while remaining computationally lightweight, making the model accessible to users with limited computing resources.

The training data consisted of manually labeled SAR images from significant flood events from the past few years. These events were chosen because we could clearly identify the flooding in SAR imagery and confirm the flooding using other sources, such as news reports, drone footage, and cloud-free Sentinel-2 imagery when available. Additionally, these flood events span multiple continents with diverse geographies, providing a robust dataset for training our model.

- Pakistan flooding in August 2022
- Greece flooding in September 2023
- Mozambique flooding in March 2023 (validation)
- Southeast Ethiopia flooding in November 2023 (test scene)

We opted for a change detection model because it leverages the temporal differences in SAR amplitudes to accurately represent flooding. In essence, by comparing SAR images captured before and after a flood event, the model can detect changes in the backscatter signal that indicate the presence of water in previously dry areas. Through experimentation on our validation dataset, we found that model performance improved when we explicitly applied filtering to the SAR amplitudes to identify ranges of pixel values consistent with the presence of water, rather than using raw amplitudes directly.

Our model uses four input features that capture both the presence of water-like signatures and significant changes in surface characteristics: binary change indicators for VV and VH polarizations (indicating transitions into typical water backscatter ranges), and the magnitude of backscatter changes (delta amplitudes) in both bands. Water typically appears dark in SAR imagery due to specular reflection, with characteristic backscatter values below −17.5 dB in the VV band and below −22.5 dB in the VH band. The delta amplitude features allow the model to distinguish between small changes that might be due to noise versus larger changes more likely to indicate actual flooding.

The Sentinel-1 SAR data used in this study is from Catalyst (via the Microsoft Planetary Computer) and has undergone standard SAR preprocessing, including orbit application, gamma-nought correction to normalize backscatter across different incidence angles, radiometric terrain correction using PlanetDEM, and speckle filtering. The gamma-nought correction adds an additional cosine correction to better normalize backscatter across different incidence angles. We utilize both ascending and descending passes to maximize temporal coverage. The model was trained on consecutive pairs of SAR images with the same viewing geometry and time of day taken within 30 days of each other. While Sentinel-1 typically has a 12-day repeat cycle, we allow up to 30 days between observations to account for potential missing acquisitions while maintaining consistent imaging conditions. Prior work[17] has suggested that including an additional pre-event image can provide some improvement in flood detection, but the gains in F1 score were small (typically less than 1 percentage point) and were often confounded with model architecture changes. Given these modest gains and the increased computational cost of processing additional images, we opted for a paired-image approach, using one pre-event and one post-event image.

While we tested incorporating additional inputs such as soil moisture and elevation data directly into the model, our labeled flood examples did not contain enough diversity across different soil moisture conditions and terrain types to effectively train on these variables. Namely, we did not see sufficient flooded and non-flooded pixels across a range of different soil moisture and slope values. Additionally, the coarse resolution of soil moisture data (10 km) meant we essentially had one measurement per scene, providing insufficient data for the model to learn meaningful relationships. We therefore elected to use these auxiliary datasets in post-processing instead, where they help reduce false positives through heuristic filtering.

### Model validation

We validated the model using two approaches: evaluation on our internal test set and comparison to the Kuro Siwo dataset, a newly-released comprehensive global dataset of expert-annotated Sentinel-1 flood images[17].

The model's performance on the test set showed promising results. To understand these results, we use several standard metrics: the Intersection over Union (IOU) measures how well our predicted flood areas overlap with actual flood areas, with our score of 0.67 meaning that 67% of the combined predicted and actual flood area was

**Table 3 | Model Validation against Kuro Siwo global dataset**

| Model | Prec | Rec | F1 | IOU |
|---|---|---|---|---|
| AI4G Model | 0.84 | 0.72 | 0.77 | 0.63 |
| GFM | 0.73 | 0.70 | 0.72 | 0.56 |
| Kuro Siwo baseline | | | 0.75–0.80 | |

Model performance against the test set in the Kuro Siwo dataset. Our model performance exceeds that of the Global Flood Monitoring (GFM) data and is on par with the models trained on the Kuro Siwo dataset itself.

correctly identified. The Precision score of 0.68 indicates that when our model predicts a flood, it's correct 68% of the time. Our Recall score of 0.99 means we successfully detected 99% of all actual flood events. Finally, the F1 Score of 0.80 represents the overall balance between precision and recall, where 1.0 would be perfect performance. These metrics indicate that the model is highly effective at detecting flood events, with a high recall ensuring most flood events are captured, albeit with some false positives, or pixels predicted as flooded but that are non-flooded.

To ensure robustness, we also test the model's performance against the Kuro Siwo dataset. This dataset was selected due to multiple favorable factors. First, it contains both pre- and post-flood SAR imagery, making it suitable for a change detection model such as ours. Second, it contains 43 distinct flood events distributed across six continents and varied climate zones. This is essential for validating a model that is being applied globally. Lastly, the dataset contains manually annotated images to account for shortcomings in existing annotation approaches such as the Copernicus Emergency Management System outputs[17]. Given the known challenges of SAR in complex terrains like steep mountains and dense forests—areas where our model's predictions are masked using the exclusion mask (see "Methods" subsection on Post-Processing)—this validation focuses on assessing model performance in the diverse global regions where detections are expected to be most reliable.

Our model performs well against the Kuro Siwo dataset, achieving an F1 score on the test set of 0.77. This is comparable to the models published in the Kuro Siwo paper, which have F1 scores ranging from 0.75 to 0.80, and greater than the performance of the Copernicus GFM system's F1 score of 0.72, as shown in Table 3. We note that in the Kuro Siwo paper they report results for multi-class classification models, so that the F1 scores are not necessarily directly comparable to the task we have for binary classification. Additionally, the paper only provides F1 scores for the flooding task. Nevertheless, their numbers provide a useful benchmark for comparison. We also note that while our model provides binary flood/no-flood classifications, GFM produces probabilistic flood likelihood values that can be thresholded for different applications. For this comparison, we used an optimal threshold of 0.3 for GFM, determined through validation on a subset of flood events (see SI Section S1.5.2).

More details on the comparisons, commentary on specific scenes in the Kuro Siwo test set, and the choice of buffer can be found in Supplementary Information Section S1.5.

## Post-processing
There are two types of post-processing we apply to the model results. First, we filter out potential false positives using both time-varying auxiliary data sources and static features. Second, we provide a static exclusion mask that identifies areas where flood detection may be unreliable due to false negatives or false positives.

There are several factors that can result in false positives in SAR data. In SAR imagery, water appears very dark. Any surface that appears similarly dark can therefore mimic water and therefore flooding. For example, arid regions often look dark in SAR because the surface absorbs much of the SAR signal or reflects it away from the

satellite, resulting in low backscatter measured by the satellite. Another cause of false positives is mountainous areas, where terrain shadows can result in areas where the signal sent out by the SAR satellite does not reach the ground, again resulting in a low SAR amplitude. Similar effects occur in heavily vegetated areas and urban areas. Lastly, freeze/thaw cycles can result in areas that technically flood, in the sense that water is present where it was not present before, but typically this flooding is of a different nature than most of the flood events we are concerned with, and so we aim to filter those out as false positives.

To address potential false positives, we incorporated both static landscape features and time-varying characteristics. The static filtering was applied based on the ESA land cover mapping[33] and a digital elevation model[34]. Specifically, we filter out areas marked as "Bare Ground" in the land cover mapping along with areas with terrain slopes greater than 10°. We include two time-varying features: soil moisture estimates[35–37] and land surface temperature[38], to identify areas with low soil moisture or low temperatures to exclude. As noted above, while we tested incorporating soil moisture and elevation data directly into the model training, we found it more effective to use it as a post-processing filter to remove false positives. We also used the ESA land cover mapping to remove permanent water bodies like lakes and rivers. While our change detection approach should inherently ignore permanent water bodies since they appear as water in both pre- and post-event imagery, we found that explicitly masking them helped reduce noise in our predictions.

We first run the neural network model to generate potential candidates for detected floods, then apply heuristic thresholds with the auxiliary datasets to remove false positives. The heuristic thresholds were determined by analyzing model predictions and auxiliary dataset values in regions with a very low likelihood of flooding, such as deserts and mountainous areas. For instance, regions with soil moisture levels below a certain threshold were excluded as false positives. We note that in the dataset we have provided, we include all potential flood candidates with the auxiliary data included, enabling other researchers to apply their own thresholds and make informed decisions on which events to filter out. Detailed methods and the specific thresholds for removing false positives are provided in Supplementary Information Section S2.

For our static map products (GeoTIFF format), we include an exclusion mask that serves two purposes. First, it identifies areas where we expect unreliable flood detection due to static landscape features based on the filtering used to minimize false positives: areas marked as "bare ground" in the ESA land cover mapping and regions with steep terrain (slopes exceeding 10°) in their immediate surroundings. We include these surrounding areas because steep terrain can influence radar signals in nearby pixels. Second, the mask identifies areas where we expect a high rate of false negatives, specifically areas marked as "Built-Up" in the ESA land cover map, which typically indicates urban areas where building interference makes flood detection challenging.

## Flooding trends over time
We analyzed flooding trends over a decade by aggregating flood extent data by month and normalizing by the number of available SAR observations. This normalization is necessary as observation frequency can vary - for example, one Sentinel-1 satellite went offline in late 2021, reducing observations by nearly 50%. To estimate changes in flooding over time, we fit a linear model to the time series under various scenarios. We chose a linear model for its simplicity and interpretability, given the noisy nature of the flood extent data and the relatively short time period.

We explored different scenarios to address potential data challenges. For example, 2022 showed much higher than average flood extent per observation, likely due to large flood events like in Pakistan. Given the short time period, these outliers could skew our results

towards artificially high estimates of increased flooding over time. Another challenge was in the earlier Sentinel-1 observations. Many of these used only one polarization channel, and therefore had different rates of flood detection than the typical two-channel observations that are ubiquitous after June 2017. We can correct for these differences, or remove those earlier years. Both sets of results are presented as different scenarios in Table 2. Detailed methodology for estimating flooding trends over time is provided in Supplementary Information Section S3.

### Buffering flood detections

One challenge inherent in high-resolution flood modeling using SAR imagery is the precise delineation of flood extents. While SAR imagery is effective in penetrating cloud cover and providing continuous monitoring, it may not always capture every pixel of a flooded area accurately. Factors such as SAR signal scattering, speckle noise, and surface roughness variations can result in imperfect flood detection, leading to potential underestimation of the flood extent[39–42].

To mitigate these limitations, we apply a buffering or dilation to the flood detections[39,41,42]. This process involves expanding the detected flood pixels by a specified distance to account for potential inaccuracies in the SAR data. The buffer helps include areas likely at risk of flooding that may not have been explicitly detected by the model. Buffering is particularly important in urban areas or vegetated areas (such as cropland), where buildings and vegetation can prevent the SAR signal from reaching the ground, making it difficult to detect flooding in some cases. For the geotiffs in our public dataset, we apply two versions, with buffers of 240 and 80 m. The 240 m (12 pixels) buffer ensures a conservative estimate, in the sense that we err on the side of marking something as flooded. For getting accurate estimates of cropland affects by flooding, we found that 80 m (4 pixels) gave us the closest alignment with other sources of flood extent (see Supplementary Information for details). We note that since we provide the raw model outputs in our dataset, other researchers have the flexibility to apply their own choice of buffer distances based on their specific needs.

### Basemap data

Basemaps providing geographic context for analysis and visualization of flood extent maps utilized administrative boundaries from the GADM database (Version 4.1)[43]. General map features, such as coastlines, lakes, and roads, were incorporated using data from Natural Earth (naturalearthdata.com).

### Supplementary information

Detailed descriptions of the neural network architecture, training data, model validation, post-processing steps, and the methodology used to analyze flooding trends over time are provided in the Supplementary Information.

## Data availability

The Sentinel-1 SAR input imagery used for analysis is publicly available from the Copernicus Data Space Ecosystem and were retrieved via the Microsoft Planetary Computer (https://planetarycomputer.microsoft.com/dataset/sentinel-1-rtc)[44]. The primary datasets generated and analyzed during the current study are publicly available without restriction. Specifically, the per-detection model predictions (including coordinates and auxiliary data) in Parquet format and the aggregated flood maps in GeoTIFF format, representing the minimum dataset necessary to interpret, verify and extend the main findings, have been deposited in the Hugging Face repository at: https://huggingface.co/datasets/ai-for-good-lab/ai4g-flood-dataset[45]. Geotiffs for the maps in Figs. 1–5 are available in the Hugging Face repository. Source data underlying Figs. 6 and 7 are provided as a Source Data file with this paper.

## Code availability

The Python code used for flood detection inference, along with the trained model artifact, is publicly available in the GitHub repository at: https://github.com/microsoft/ai4g-flood[46]. Documentation and instructions for running the model using the provided artifact are available within the repository.

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

## Acknowledgements
We thank the IOM Ethiopia team for their collaboration on identifying communities at risk of flooding in Ethiopia using the data presented herein. We thank Luana Marotti and Amy Michaels for their assistance with compliance and data approval requests.

## Author contributions
A.M. conceived and led the research, trained and ran inference on the model and was primarily responsible for data analysis and manuscript writing. K.W. contributed to project conceptualization, result interpretation and manuscript framing. S.N. provided critical support in model training and architectural design. W.S. offered expert guidance on flood datasets, assisted with model validation strategies and contributed to manuscript preparation. J.L. established the overall project objectives and provided leadership as the Lab Director.

## Competing interests
The authors declare no competing interests.
