## [Peer Review file · Nature Communications]

Mapping Global Floods with 10 Years of Satellite Radar Data

Corresponding Author: Dr Amit Misra

Version 0:

Reviewer comments:

Reviewer #1

(Remarks to the Author)

Mapping Global Floods with 10 Years of Satellite Radar Data

I enjoyed reading this manuscript and congratulations on doing this huge amount of important work. However, with important work comes great responsibility. If this gets published in Nature Communications, and with the Microsoft association, it will be widely picked up. It is vital to properly assess the capabilities of the data and make crystal clear the points the data can be used for and what it perhaps should not be used for or where users should be mindful. I have been involved in disaster response around flooding for a few years now and I see data, especially satellite derived data misused frequently. Users tend to view the data as the true flood which of course is nonsense as the image is only a representation of an image. Making sure this is well understood is crucial and indeed ethical.

With this in mind, I think if this data is going to be in Nature Communications and likely widely publicised, more careful evaluation and communication is needed. I have outlined my considerations below. Please do not take these observations to heart too much as I really believe you have done some great work. A few years ago I was in a similar position being solely a data creator and not really considering the implications of incorrect use in the real world, but after a few years of seeing and working in more applications, I have switched my view a little. You could easily publish this in a more specialised journal, but for a journal with so much clout I cannot emphasise enough how important it is to be robust.

Good points:

- The multi-temporality is really interesting
- A truly amazing amount of data processing
- Buffering is a good addition
- The consideration of false positives is quite robust
- The data record is nice

Major points to consider

- The figures can be very difficult to read. For instance Figure 5 is unreadable.
- Is the calculation in Figure 7 correct? Or more accurately, should a different threshold be used to determine what is significant or not? Assuming a 1 degree tile is 111km, the area of a 3x3 degree tile would be 110,889km². One hectare is 0.01km², so a large increase would be 0.2km². This is a really small fraction of the total area. I wonder would it be better to report the % total land area flooded and any trends in that. That would also get over the issue of unequal areas in each 3x3 degree tiles. You have a really nice story here, but I feel it could be told a little clearer.
- Paragraph starting at Line 232. You make some important claims in this paragraph, but the analysis for it is not in the main text. Many people use Landsat/MODIS data so this analysis should be supported adequately.
- I would highly recommend including an exclusion layer like Copernicus Global Flood Monitoring does. You correctly mention the difficulties in recording flooding in difficult areas like urban areas, but it is important to exclude these (or at least flag the inability in these areas) in your data products. Many potential users may not quite grasp this in my experience so having an exclusion layer is very useful.
- The model validation section could be improved. I would love to see some maps for each location to better understand what is really going on. What type of flood is being detected? If it is a fast onset (say pluvial), then the differences in timing of the high resolution optical imagery with the SAR imagery could be quite challenging. Could you also confirm the overpass timing of each imagery so the reader can get a better idea of the timing so we know if like for like is being compared?
- I believe you are comparing against the Global Flood Monitoring product (GFS, ref 14). Did you use the likelihood values? I have found in model evaluation and forecasting work the likelihood maps to be very useful in visualising how sensitive an

area can be to the different SAR processing algorithms. If you evaluate against different thresholds of likelihood that might produce some interesting results. It would be fantastic if your product could have some sort of uncertainty/ensemble attached to it. I have no affiliation to this group whatsoever by the way, but in my experience I have found it really useful. I note the great work in the post-processing to remove false positives as I have read on. Well done!

- As you noted, there is a lack of flood events used in the training of your model. Even if it might not improve your metrics too much, it would be very beneficial to make your work more convincing if you had more training image. I note you discuss this more in supplementary materials. I can see that 4 locations may suffice, but there others may need more convincing. How about labelled images from (from a quick google):

o Montello, F., Arnaudo, E., & Rossi, C. (2022). MMFlood: A Multimodal Dataset for Flood Delineation from Satellite Imagery (1.0.0) [Data set]. Zenodo.

- If you cannot use more images to train, it would be great to describe why the areas you used are enough for the model to paint an accurate and robust global picture.

- Again you have rightfully noted there is a lack of validation/evaluation. It is vital to note that your confidence in your results are less in places that you have not trained or are not similar. This product could well be used in a disaster situation and it is vital (and ethical) to make people aware of what the product can achieve and roughly how confident you are in it.

- Could you give some further explanation on you chose the ESRI landcover product. Is it the multi-temporality (I'm assuming it's the Dynamic world one). Or are you using a static landcover mask? Does cropland contain irrigated areas as that could significantly impact the flood results (irrigated areas, particularly in your Pakistan example are detected as flooded. They are flooded, BUT are deliberately flooded.

- I think it would be beneficial to describe why a MobileNet architecture is appropriate for this analysis. By this I mean how it processes and classifies images. I note you discuss the performance compare to ResNet etc., but for someone not familiar with ML/Deep learning it would be good to briefly describe why it is appropriate.

- Did you test the local thresholding for the SAR imagery? In my experience the thresholding is so important and what works in the floodplain of the Ganges can be wildly different to that in the Zambezi. You could effectively create ensemble maps by choosing different thresholds. I do later note the super large 2 month compute time so that may not be possible. That seems super long to me.

- Delving a bit more into the model validation in supplementary materials. The time difference between the Planet and SAR images is quite significant, especially for a relatively short flood. For a long lasting flood where the flood waters are fairly static that might suffice, but for this relatively short flood you need to highlight more the potential issues. Did you get any on the ground photos, or try to use gauge data or something like that to unpick the flood event? Or any ground reports?

- I understand why you used the CNN derived Planet delineation, but it is difficult to be convinced whether this can be truly trusted. It is tempting to believe a very fine resolution image because it looks crisper, but it can be very wrong.

- It would be good to compare your extents visually to the Pekel Global surface water, both quantitatively and visually
Minor

- There should be a mention that SAR imagery most likely misses the flood peak

- Figure 4 – Needs a legend on the actual figure itself

- Figure 6 uses square meters as the unit of analysis but others use hectares. Is this deliberate? If so why was this choice made?

- Line 230 – I would be tempted to just say other optical flood products and not constrain it to NOAA. There's a lot out there and you don't want to risk annoying anyone.

- Line 382. It would be good to describe what each metric means so a non expert could understand what a good score is. Perhaps compare these scores to something similar. Basically just make it clear what they really mean.

- Line 426 – why did you chose a linear model?

- Perhaps clarify the resolution of the product. I see on the data record it is 30m for the rest of the world, but you mention 20m in the text. When describing the buffer you are using multiple of 30m.

- Line 40 supplementary materials – what are the other sources you are describing?

- Can you run the model at different locations in different areas? In large floodplains, 20m or coarser would more than suffice. But in smaller areas/peri-urban you could run at 10m?

- I think model evaluation is better than model validation in this context as we cannot be that confident in the 'ground truth'/benchmark data.

- Figure S2 – the maps could be zoomed in more to see the areas of agreement etc. It is impossible to tell at this scale what is going on.

- Instead of slope, you could use a slope roughness metric. Check out the vast range in WhiteBox Tools

(Remarks on code availability)

The code and data record are well constructed and documented.

Reviewer #2

(Remarks to the Author)

General Comments:

The authors utilized deep learning (DL) for global flood extent mapping using 10-meter resolution SAR imagery. While the method shows potential, the study's test cases are insufficient for claiming a global-scale applicability. Additionally, there are several areas where the methodology, discussion, and framing of results can be improved for scientific rigor and clarity.

1. Clarity and Wording Issues:

• Line 9: The phrase "any weather condition" is misleading. SAR can penetrate clouds and operate both day and night, but it has limitations in rainy conditions due to rain droplets affecting backscatter. Please rephrase this line for accuracy.

- Line 35: The term "regardless of weather condition" is not accurate. SAR operates day and night and can function under cloudy conditions, but rainy weather can impact backscatter values.
- Lines 29–38: The wording in this section needs revision for better clarity and flow.
- Lines 35–36: Please clarify what is meant by this line. Are you referring to SAR's temporal resolution? If so, specify how it relates to flood mapping.
- Line 66: Rephrase this sentence for improved readability and accuracy.

2. Missing Points in the Introduction:

The introduction lacks critical context regarding SAR capabilities and related studies. I recommend addressing the following:

- The capability of SAR for water mapping and flood detection, emphasizing its ability to detect dynamic water changes.
- A brief literature review on prior studies that used SAR (especially Sentinel-1) for water and flood mapping.
- The rationale for using only SAR imagery. Why was SAR-optical fusion not considered? Optical imagery, while affected by cloud cover, can complement SAR and improve flood mapping accuracy.
- The number of Sentinel-1 images required before and after a flood event to ensure reliable analysis.

3. SAR Data and Processing:

- Lines 404–421: The discussion on SAR imagery is not scientifically sound. Key points to address:
 - o Include the range of SAR backscatter values typically associated with water.
 - o Discuss the need for incidence angle correction, as this is critical for comparing backscatter across different tiles. The incidence angle could also be included as an input feature in the model.
 - o Explain why soil moisture data was not considered as an input feature to improve model performance. Soil moisture can significantly impact backscatter and flood detection.
- Preprocessing:
 - o Specify the preprocessing techniques applied to the 10-meter SAR data. Given the resolution, speckle noise is a significant issue, and speckle filtering is essential. Clarify if and how this was handled.
 - o Did the study use ascending and descending passes, or only one of them? This should be stated explicitly.
- Feature Importance:
 - o Consider performing a feature importance analysis to identify which auxiliary data contributes most to model performance. This would help reduce redundancy and improve the model's efficiency. Refer to studies that demonstrate the importance of feature selection in deep learning.
 - o Specifically, it is necessary to determine the most important SAR features for flood mapping. For example, features like backscatter intensity, polarization (VV/VH), or temporal change metrics may vary in their significance depending on the region and type of flooding.

o This approach could also guide the integration of auxiliary data, ensuring only the most impactful variables are included in the model. Refer to existing studies that highlight the benefits of feature selection in deep learning models.

- Reliability of Backscatter:
 - o Acknowledge the limitations of SAR data on rainy days. Rain droplets and dew can alter backscatter values and impact classification accuracy. This should be explicitly discussed.

4. Differentiating Between Water Types:

- Provide an explanation of how the model differentiates:
 - o Permanent surface water versus flooded areas.
 - o Flooded cropland (e.g., rice paddies in Southeast Asia) versus actual flood events.
- Special test cases should target regions prone to misclassification, such as paddy fields or other seasonally flooded croplands. For example, line 105 mentions that 11% of cropland lies in historically flooded areas—these regions need focused testing.

5. Comparison and Validation:

- It would strengthen the study to compare the performance of:
 - o SAR-only models.
 - o Optical-only models.
 - o Fusion of SAR and optical data. This would provide insights into the complementary nature of these datasets for flood mapping.
- Discuss SAR's unique ability to detect dynamic water changes and compare it with prior studies.

6. Testing and Evaluation:

- The study needs to expand its testing regions, particularly targeting areas with complex hydrological dynamics (e.g., Southeast Asia, which has flooded cropland like rice fields). The current test cases are insufficient to generalize the findings to a global scale.

(Remarks on code availability)

Reviewer #3

(Remarks to the Author)

This paper describes the use of a ten-year record of Sentinel-1 SAR images to map floods globally, using a neural network along auxiliary dataset, including soil moisture, DEM, temperature, and land cover to minimize false positive errors.

It demonstrates trends at the global scale and assess some use cases around the world where the authors also validate the maps with high-resolution optical images from Planet Labs.

The paper is generally very well written and the methodology is sound, both technically and statistically.

In the main text of the paper, an in-depth comparison with and discussion of the Copernicus GFM S1 based flood maps, especially on the use cases presented would be much appreciated by a larger audience, I think. In my opinion, the authors should add this and also describe better the benefits and advantages/limitations of their global method compared to the Copernicus GFM.

In the Supplementary material. IOU (F1) is indeed low, maybe the authors should have some cases with S2 as a validation set and discuss the differences and thus maybe prove their point about the difficulty with high res optical images as a validation set.

(Remarks on code availability)

I saw no code being shared. My apologies if I missed this.

Version 1:

Reviewer comments:

Reviewer #1

(Remarks to the Author)

Well done to the authors for the pain-staking response to all the reviewers comments. I particularly appreciate the additional validation and inclusion of a exclusion mask.

The improvement to the Figures are appreciated. Personally I would like to see a scale bar on some of the more detailed map, but that's just being cartographically picky!

I find the large increase in flood detection from your method very intriguing!

It would be very interesting to evaluate your approach with this new Bayesian based work from Roth et al (2025) but that is probably out of the scope of this paper

Roth, F., Tupas, M.E., Navacchi, C., Zhao, J., Wagner, W. and Bauer-Marschallinger, B., 2025. Evaluating the robustness of Bayesian flood mapping with Sentinel-1 data: A multi-event validation study. *Science of Remote Sensing*, 11, p.100210.

(Remarks on code availability)

The code is hosted on github and is adequately documented

Reviewer #2

(Remarks to the Author)

Dear Authors,

I appreciate your effort in compiling a global flood extent dataset and acknowledge the importance of large-scale flood mapping initiatives. However, I would like to offer several comments and suggestions that I believe can help improve the clarity, rigor, and overall impact of your manuscript.

1. Limited Case-Based Evaluation One key concern is the generalization of algorithm performance based on a limited number of flood cases. It is difficult to conclude whether one algorithm is superior to another without extensive and systematic validation across diverse flood events and environmental conditions. Many similar studies fall short in offering insights into the strengths and limitations of individual algorithms, making it challenging for readers to understand their applicability across different geographies and weather conditions.

I would strongly encourage the authors—and our research community at large—to place more emphasis on rigorous validation and physical understanding, rather than solely on developing increasingly complex algorithms. In this regard, I would like to highlight the exemplary work by Florian Roth et al. (2025), who conducted a robust multi-event evaluation of a Bayesian flood mapping algorithm using 18 flood events across five continents with Sentinel-1 data. His study offers critical insights into the sensitivity of VV and VH polarizations over vegetated flood areas and identifies important limitations in existing approaches. This paper is currently missing from your references and I highly recommend reviewing it:

Roth, F., Tupas, M.E., Navacchi, C., Zhao, J., Wagner, W., Bauer-Marschallinger, B., 2025. Evaluating the robustness of Bayesian flood mapping with Sentinel-1 data: A multi-event validation study. *Science of Remote Sensing*, 100210. Link
Notably, this work has already informed improvements in the operational Copernicus Global Flood Monitoring (GFM) service.

2. Motivation for Temporal Aggregation While aggregating 10 years of flood extent data may be useful from a historical perspective, the added value is not fully clear in your current discussion. What does this aggregated product offer, and how can it be linked to broader trends such as climate change, land use changes, or other hydrometeorological drivers? Without such context, it is difficult to interpret the observed increase or decrease in flood extents across regions.

3. Specific Comments on the Manuscript

- Lines 33–36: This sentence needs rephrasing for clarity and to avoid redundancy.

- Lines 39–40: The claim that "daily observations from constellations" are available is incorrect. For HLS (Harmonized Landsat-Sentinel), the typical revisit time is 2–3 days, not daily.

- Lines 66–67: Please clarify the specific benefit of the proposed approach in this context.
 - Line 72: What is the practical advantage of creating a 10-year SAR-based flood extent database, especially considering Sentinel-1's limitations (e.g., revisit time, difficulty with forested flood areas)?
 - Line 83–84: While historical flooding can indicate areas of potential future risk, it is important to acknowledge the role of climate change and other dynamic factors that may alter future flood patterns. This aspect is missing from your discussion.
4. Issues with the Introduction and Figures
- The introduction could be improved for clarity and conciseness. The last paragraph, in particular, repeats earlier points and would benefit from restructuring.
 - Figure 1: The color scheme and visual quality are suboptimal. The figure is difficult to interpret, and the visualization should be improved.
 - The masking of regions (due to urbanization, desert areas, etc.) appears to have excluded countries like Iran, which have experienced significant flood events. Please explain the rationale and implications of this exclusion.
 - The validation approach remains unclear, particularly for flood-prone areas such as mountainous regions and flooded forests, which are often excluded or poorly captured.
 - Lines 153–163: The justification provided for why your approach captures "additional" flood areas more accurately than previous methods is not convincing. More evidence or validation is needed.
 - Figure 8: To enhance its value, consider linking it to climate-related drivers of flood change. Without this context, the figure's contribution to advancing flood mapping knowledge remains limited.

(Remarks on code availability)

Reviewer #3

(Remarks to the Author)

I have read the responses to comments and the revised version, and I thank the authors for their extensive revisions.

In my opinion, the responses and revisions with regard to my points of concern raised are adequate and sufficient.

(Remarks on code availability)

I have not reviewed the code. I do not feel in the best position to review computer code in the sense of bugs and inconsistencies.

Version 2:

Reviewer comments:

Reviewer #2

(Remarks to the Author)

I would like to thank the authors for their thoughtful responses to my previous comments. I appreciate the effort they have put into revising the manuscript and addressing the concerns raised.

That said, for several of the comments, I had expected more substantial revisions—particularly beyond citing prior work or adding a few lines in the introduction or discussion. With all due respect to the authors' efforts and the use of an extensive 10-year Sentinel-1 dataset, I am unable to identify a clear added value or novel contribution that significantly advances the field or benefits future research. As such, I remain unconvinced of the broader impact of this study on the SAR flood mapping community.

Although I have no further technical comments at this stage, I do not believe the manuscript meets the threshold for publication in its current form. Given the authors' considerable efforts, I defer the final decision to the editor and would also welcome input from an additional reviewer for a broader perspective.

(Remarks on code availability)

REVIEWER COMMENTS

Reviewer #1 (Remarks to the Author):

Mapping Global Floods with 10 Years of Satellite Radar Data

I enjoyed reading this manuscript and congratulations on doing this huge amount of important work.

However, with important work comes great responsibility. If this gets published in Nature Communications, and with the Microsoft association, it will be widely picked up. It is vital to properly assess the capabilities of the data and make crystal clear the points the data can be used for and what it perhaps should not be used for or where users should be mindful. I have been involved in disaster response around flooding for a few years now and I see data, especially satellite derived data misused frequently. Users tend to view the data as the true flood which of course is nonsense as the image is only a representation of an image. Making sure this is well understood is crucial and indeed ethical.

With this in mind, I think if this data is going to be in Nature Communications and likely widely publicised, more careful evaluation and communication is needed. I have outlined my considerations below. Please do not take these observations to heart too much as I really believe you have done some great work. A few years ago I was in a similar position being solely a data creator and not really considering the implications of incorrect use in the real world, but after a few years of seeing and working in more applications, I have switched my view a little. You could easily publish this in a more specialised journal, but for a journal with so much clout I cannot emphasise enough how important it is to be robust.

Good points:

- The multi-temporality is really interesting
- A truly amazing amount of data processing
- Buffering is a good addition
- The consideration of false positives is quite robust
- The data record is nice

Response: We thank the reviewer for their kind words and their feedback. Point well taken on the need to more careful and deliberate in evaluation and communication of the results. We believe we have addressed this by making the suggestions made below, specifically by doing much more extensive validation of the model against other datasets and including an exclusion mask in our maps to show areas where the model results should be interpreted with caution.

Major points to consider

- The figures can be very difficult to read. For instance Figure 5 is unreadable.

Response: We've redone many of the figures to make them more legible.

- Is the calculation in Figure 7 correct? Or more accurately, should a different threshold be used to determine what is significant or not? Assuming a 1 degree tile is 111km, the area of a 3x3 degree tile would be 110,889km². One hectare is 0.01km², so a large increase would be

0.2km². This is a really small fraction of the total area. I wonder would it be better to report the % total land area flooded and any trends in that. That would also get over the issue of unequal areas in each 3x3 degree tiles. You have a really nice story here, but I feel it could be told a little clearer.

Response: This is a good suggestion – we’ve converted this to changes in percentage of land area flooded and added a filter on the magnitude of the change so that we don’t plot very small absolute changes that are large in percentages. To your point about the magnitude of the trend, there was initially an error in the calculation because we did not properly account for the dilation/buffering, but your overall point still stands – these are not huge increases relative to the total land area in each tile. In fact, most tiles are <<1% flooded on average. Overall the plot looks very similar, but we believe that following your suggestion will help the story come across more clearly.

One note on the new plot – you’ll see that the estimated trends are relatively large – often in the 1-2% range per month, which translates to quite large increases over 10 years. Because of the limited time frame of our analysis, we typically don’t observe a steady increase over time in flooding. Instead, the trends are often dominated by a couple large flood events in the dataset, resulting in large estimated slopes. We’ve tried to capture this concept in the text and figure caption.

- Paragraph starting at Line 232. You make some important claims in this paragraph, but the analysis for it is not in the main text. Many people use Landsat/MODIS data so this analysis should be supported adequately.

Response: We've added a detailed comparison to both the Pekel Global Surface Water (GSW) and MODIS-based datasets in the Results section. We now present global estimates of overlap and potential new flood-prone regions our method identifies, along with estimates for each of the regions we call out in the text and figures. This is all in the “Global Flood Map and Application to Ethiopia” subsection under the Results section. Overall, we find that there is reasonable overlap with these datasets, but also a large amount of net new flooding detected by our model.

- I would highly recommend including an exclusion layer like Copernicus Global Flood Monitoring does. You correctly mention the difficulties in recording flooding in difficult areas like urban areas, but it is important to exclude these (or at least flag the inability in these areas) in your data products. Many potential users may not quite grasp this in my experience so having an exclusion layer is very useful.

Response: This is a great suggestion, and in the aggregate maps we provide we've added exclusion layers, similar to the approach used by Copernicus GFM. To be more specific, the exclusion mask includes areas we either expect false negatives (urban areas) or have excluded areas due to potential false positives (arid regions and rough terrain). This is shown in the new plots, along with additional text explaining this idea in the Results and Methods section

- The model validation section could be improved. I would love to see some maps for each location to better understand what is really going on. What type of flood is being detected? If it is a fast onset (say pluvial), then the differences in timing of the high resolution optical imagery

with the SAR imagery could be quite challenging. Could you also confirm the overpass timing of each imagery so the reader can get a better idea of the timing so we know if like for like is being compared?

Response: We've added maps for the new model validation approach (see later comment + response for more details). More importantly, we've shifted our primary validation to use the Kuro Siwo dataset, a comprehensive SAR-specific dataset, which eliminates the challenges of timing mismatches between SAR and optical imagery. We've included some maps from the Kuro Siwo validation in the SI to highlight areas where the model does well and not so well.

- I believe you are comparing against the Global Flood Monitoring product (GFS, ref 14). Did you use the likelihood values? I have found in model evaluation and forecasting work the likelihood maps to be very useful in visualising how sensitive an area can be to the different SAR processing algorithms. If you evaluate against different thresholds of likelihood that might produce some interesting results. It would be fantastic if your product could have some sort of uncertainty/ensemble attached to it. I have no affiliation to this group whatsoever by the way, but in my experience I have found it really useful. I note the great work in the post-processing to remove false positives as I have read on. Well done!

Response: Regarding the likelihood values, in the new model validation approach we did experiment with these and seeing how the performance changes with different thresholds. We do find that model performance can be improved by reducing the threshold, so we report results in the Supplementary Information under different scenarios. Detailed results across different thresholds are reported in the Supplementary Information Section S1.5.2

For our current model, we do not have well-calibrated uncertainties. We agree that would be a valuable thing to include, however, this is a known challenge with deep learning models. Our current model's predictions tend to be highly overconfident in either direction, which doesn't lend itself well to proper calibration of uncertainties or probabilities. It is possible to build a better calibrated model to produce uncertainties like the GFM does, but for now we leave that for future work.

- As you noted, there is a lack of flood events used in the training of your model. Even if it might not improve your metrics too much, it would be very beneficial to make your work more convincing if you had more training image. I note you discuss this more in supplementary materials. I can see that 4 locations may suffice, but there others may need more convincing. How about labelled images from (from a quick google):

o Montello, F., Arnaudo, E., & Rossi, C. (2022). MMFlood: A Multimodal Dataset for Flood Delineation from Satellite Imagery (1.0.0) [Data set]. Zenodo.

- If you cannot use more images to train, it would be great to describe why the areas you used are enough for the model to paint an accurate and robust global picture.

Response: Based on comments from multiple reviewers, we have done additional model validation. While we considered adding more training data, our validation results suggest that our current training set, though small, is sufficiently diverse and representative. We found the recently published Kuro Siwo dataset to be the best match for validating our approach, since the dataset includes pre and post SAR imagery and labels for 43 locations across the globe. We've

re-written the model validation section of the paper to reflect this new work. See “Model Validation” under the Methods section and S1.5.2 in the SI. In summary, we find that our model performs well on this dataset, showing strong average performance across continents and similar aggregate performance to what the authors report in the Kuro Siwo paper, along with slightly better performance than the GFM ensemble product.

- Again you have rightfully noted there is a lack of validation/evaluation. It is vital to note that your confidence in your results are less in places that you have not trained or are not similar. This product could well be used in a disaster situation and it is vital (and ethical) to make people aware of what the product can achieve and roughly how confident you are in it.

Response One challenge is that we don't have well calibrated uncertainties from the model. While we cannot fully address this issue in the current work, we've taken several steps to help users understand the model's limitations and reliability. Following your suggestion, we've added exclusion layers to the maps to clearly indicate areas where the model is known to be less reliable - such as arid regions and mountainous regions (prone to false positives) or urban areas (prone to false negatives). We've also added explicit discussion of these limitations in the paper's Discussion section to ensure users, particularly those in disaster response situations, are aware of where and when the model's predictions should be used with additional caution.

- Could you give some further explanation on you chose the ESRI landcover product. Is it the multi-temporality (I'm assuming it's the Dynamic world one). Or are you using a static landcover mask? Does cropland contain irrigated areas as that could significantly impact the flood results (irrigated areas, particularly in your Pakistan example are detected as flooded. They are flooded, BUT are deliberately flooded).

Response: We are using a static land cover map from ESRI. It does have a temporal component, but given that it doesn't cover the entire time period of the SAR data, we decided to just use the most recent mapping. We're using the ESRI one for this particular use case because it's been found to be more accurate for cropland than ESA world cover (see Zentner et al 2022). The cropland does contain irrigated areas, and we've added a caveat in the Discussion section about how to interpret results in irrigated cropland, particularly noting the case of rice paddies and other intentionally flooded agricultural areas.

- I think it would be beneficial to describe why a MobileNet architecture is appropriate for this analysis. By this I mean how it processes and classifies images. I note you discuss the performance compare to ResNet etc., but for someone not familiar with ML/Deep learning it would be good to briefly describe why it is appropriate.

Response: We've added this to the Methodology Section: 'We chose MobileNet as our model architecture because it can be effectively adapted for pixel-level classification tasks in satellite imagery while remaining computationally lightweight, making the model accessible to users with limited computing resources.'

- Did you test the local thresholding for the SAR imagery? In my experience the thresholding is so important and what works in the floodplain of the Ganges can be wildly different to that in the Zambezi. You could effectively create ensemble maps by choosing different thresholds. I do

later note the super large 2 month compute time so that may not be possible. That seems super long to me.

Response: We did not implement local thresholding for this work. While we agree this would be valuable, our validation against the global Kuro Siwo dataset shows that global thresholding performs well across diverse geographic regions (with F1 scores ranging from 0.68-0.80 across continents, as detailed in the SI). We've documented specific cases where global thresholding may be suboptimal in the SI, but leave the implementation of local thresholding for future work.

- Delving a bit more into the model validation in supplementary materials. The time difference between the Planet and SAR images is quite significant, especially for a relatively short flood. For a long lasting flood where the flood waters are fairly static that might suffice, but for this relatively short flood you need to highlight more the potential issues. Did you get any on the ground photos, or try to use gauge data or something like that to unpick the flood event? Or any ground reports?

Response: For this particular flood, we did not have on the ground photos. We agree that the time difference is a major issue. To help address concerns over model accuracy, we have shifted our primary validation to the new SAR-specific Kuro Siwo dataset, which shows strong performance across diverse global locations. While we've maintained the Kenya flood as a valuable real-world case study, it is no longer the primary validation of the model.

- I understand why you used the CNN derived Planet delineation, but it is difficult to be convinced whether this can be truly trusted. It is tempting to believe a very fine resolution image because it looks crisper, but it can be very wrong.

Response: This is a fair point about the limitations of fine-resolution imagery. We've addressed this concern by shifting to the Kuro Siwo SAR dataset as our primary validation source. While we retain the Planet imagery comparison as part of our Kenya case study, it is no longer used for model validation.

- It would be good to compare your extents visually to the Pekel Global surface water, both quantitatively and visually

Response: We've added plots and a table comparing our flood map to the Pekel map and the MODIS-based flood dataset globally and for the specific regions we mention in the Discussion. Overall, we find that the overlap between our model and the other datasets is consistent with how closely the two other dataset match each other – some amount of overlap but many unique flood detections in each one. This is not surprising – all three datasets cover different time periods with different observing instruments with different temporal resolution. Overall, the point we're making is that there are areas (like the ones in Ethiopia we highlight in the paper) where our model is capturing significantly more flood-prone area than the other two datasets. This is shown in Results section, specifically the subsection "Global Flood Map and Application to Ethiopia"

Minor

- There should be a mention that SAR imagery most likely misses the flood peak

Response: We've added a line on the temporal limitations of the SAR imagery in the Introduction.

- Figure 4 – Needs a legend on the actual figure itself

Response: Added

- Figure 6 uses square meters as the unit of analysis but others use hectares. Is this deliberate? If so why was this choice made?

Response: We've changed this to hectares. Also, the units in the plot were initially normalized to the number of observations in each month, since we do this for the trend analysis (to account for variations in observations per month, for example when S1B went down). The new plot is in our best estimate of hectares/month flooded

- Line 230 – I would be tempted to just say other optical flood products and not constrain it to NOAA. There's a lot out there and you don't want to risk annoying anyone.

Response: Done

- Line 382. It would be good to describe what each metric means so a non expert could understand what a good score is. Perhaps compare these scores to something similar. Basically just make it clear what they really mean.

Response: This has been added to the Model validation section when the metrics are introduced.

- Line 426 – why did you chose a linear model?

Response: We added this line to the text: "We chose a linear model for its simplicity and interpretability, given the noisy nature of the flood extent data and the relatively short time period."

- Perhaps clarify the resolution of the product. I see on the data record it is 30m for the rest of the world, but you mention 20m in the text. When describing the buffer you are using multiple of 30m.

Response: Yes you're correct – we initially ran this at 30m resolution but have been updating everything to 20m resolution. We've updated the results to be 20m resolution with buffers that are multiples of 20 and have addressed this in the text.

- Line 40 supplementary materials – what are the other sources you are describing?

Response: It's the datasets that were mentioned two paragraphs earlier: Sentinel-2 imagery, news articles, drone footage and UN reports. We've clarified this in the text.

- Can you run the model at different locations in different areas? In large floodplains, 20m or coarser would more than suffice. But in smaller areas/peri-urban you could run at 10m?

Response: We did not do this, but it's one of the options for running the model. Since the model and code are available on github, this is something other researchers could do based on their needs. We've added this point to the text.

- I think model evaluation is better than model validation in this context as we cannot be that confident in the 'ground truth'/benchmark data.

Response: Agreed – we now refer to validation only against the Kuro Siwo dataset, and discuss the Kenya analysis as an 'Evaluation' instead

- Figure S2 – the maps could be zoomed in more to see the areas of agreement etc. It is impossible to tell at this scale what is going on.

Response: For the new validation, we have some more zoomed in maps. Since the Kenya validation/evaluation is no longer central to the paper, we've left those images as is.

- Instead of slope, you could use a slope roughness metric. Check out the vast range in WhiteBox Tools

Response: Interesting suggestion, we hadn't seen this before! We'll consider this in any future work.

Reviewer #1 (Remarks on code availability):

The code and data record are well constructed and documented.

Reviewer #2 (Remarks to the Author):

General Comments:

The authors utilized deep learning (DL) for global flood extent mapping using 10-meter resolution SAR imagery. While the method shows potential, the study's test cases are insufficient for claiming a global-scale applicability. Additionally, there are several areas where the methodology, discussion, and framing of results can be improved for scientific rigor and clarity.

Response: We thank the reviewer for their helpful suggestions. Per their feedback, we have made a significant change to the Model Validation. As detailed below, we have evaluated our model against a newly released global dataset of SAR images and flood labels from 43 locations called Kuro Siwo. Our model shows good performance compared to both the Kuro Siwo baseline in their paper and to Copernicus GFM model ensemble predictions.

We have also revised and added much of the text to address comments on scientific rigor and clarity. Details of our responses to the individual comments are below

1. Clarity and Wording Issues:

- Line 9: The phrase "any weather condition" is misleading. SAR can penetrate clouds and operate both day and night, but it has limitations in rainy conditions due to rain droplets affecting backscatter. Please rephrase this line for accuracy.
- Line 35: The term "regardless of weather condition" is not accurate. SAR operates day and night and can function under cloudy conditions, but rainy weather can impact backscatter values.

Response: Thank you for this important technical clarification. We have modified the text to remove phrases like 'any weather condition' and instead focus on SAR's cloud-penetrating capabilities and day/night operation. We have added appropriate citations acknowledging the potential impact of extremely heavy rainfall on SAR backscatter in the introduction.

- Lines 29–38: The wording in this section needs revision for better clarity and flow.

Response: We have substantially revised this paragraph to improve clarity and flow. The new text provides a more structured comparison of optical/infrared and SAR technologies, with precise details about their respective capabilities and limitations. We have also added specific information about temporal coverage (e.g., 6-12 days for Sentinel-1 versus daily for some optical satellites) and clearly articulated the trade-offs between the different satellite systems.

Lines 35–36: Please clarify what is meant by this line. Are you referring to SAR’s temporal resolution? If so, specify how it relates to flood mapping.

Response: Yes we are referring to the temporal resolution. We have clarified how the 6-12 day revisit period for a satellite constellation like Sentinel-1 can affect our ability to fully capture the dynamics of flooding, particularly for short-lived events such as flash floods.

- Line 66: Rephrase this sentence for improved readability and accuracy.

Response: We’ve changed this to: “Developing a comprehensive, high-resolution map that identifies all locations where floods were detected globally over the past decade” to clarify what we mean and provide a more accurate statement

2. Missing Points in the Introduction:

The introduction lacks critical context regarding SAR capabilities and related studies. I recommend addressing the following:

- The capability of SAR for water mapping and flood detection, emphasizing its ability to detect dynamic water changes.
- A brief literature review on prior studies that used SAR (especially Sentinel-1) for water and flood mapping.

Response: These points are now addressed in the introduction, where we discuss SAR’s capabilities for water mapping, particularly its distinct backscatter signature due to specular reflection. Given space constraints, we have focused on citing a comprehensive review of SAR flood mapping techniques (from threshold-based approaches to deep learning) to cover the basics of how SAR is used for water and flood mapping.

- The rationale for using only SAR imagery. Why was SAR-optical fusion not considered? Optical imagery, while affected by cloud cover, can complement SAR and improve flood mapping accuracy.

Response: We have clarified in the text our rationale for focusing solely on SAR imagery in the introduction stating “We focus solely on SAR to ensure consistent detection through cloud cover and in both day and night conditions, which is crucial for creating reliable aggregate flood maps and enabling unbiased temporal analysis.”

Additionally, since previous work has created aggregate flood maps using optical data (from MODIS and Landsat), we view our work with SAR as being a novel contribution to the field.

Lastly, we have added fusion techniques as a potential future research direction and now include a paragraph on this in the Discussion.

- The number of Sentinel-1 images required before and after a flood event to ensure reliable analysis.

In our methodology, we use pairs of consecutive SAR images taken within 30 days of each other. Recent results from the Kuro Siwo paper (Bountos et al., 2024) indicate that while incorporating an additional pre-event image can provide some improvement, the gains in F1 score are small (typically less than 1 percentage point) and often confounded with changes in model architecture. Given these modest benefits and the increased computational cost, a paired-image approach is appropriate. We have added this explanation to the Methods section. While time-series approaches, such as those used in the Sen12 dataset, may offer further improvements in flood classification, their significantly higher computational requirements made them out of scope for our work.

3. SAR Data and Processing:

- Lines 404–421: The discussion on SAR imagery is not scientifically sound. Key points to address:

- o Include the range of SAR backscatter values typically associated with water.

Response: We have added this to the Methods section. Specifically, we identified characteristic backscatter values below -17.5 dB in the VV band and below -22.5 dB in the VH band as typical for water surfaces due to specular reflection

- o Discuss the need for incidence angle correction, as this is critical for comparing backscatter across different tiles. The incidence angle could also be included as an input feature in the model.

Response: Agreed, and we've added details on this to the text. We're using data that is gamma-nought corrected, which includes an extra cosine correction to better normalize backscatter across different incidence angles. Including incidence angle as a feature is a good point that we had not considered – this is a great idea for future work.

- o Explain why soil moisture data was not considered as an input feature to improve model performance. Soil moisture can significantly impact backscatter and flood detection.

Response: Yes, we agree on the importance of soil moisture. We actually did test this approach, as briefly noted in the text. However, given your interest in this point, we have expanded this text in the main paper under the 'Neural Network Model' subsection. In short, because of the coarse resolution for soil moisture (~10km), we did not have sufficient diversity in flood and non-flood events across different soil moistures to adequately learn the relationships between soil moisture and flooding in the model, and therefore opted to include soil moisture and slope data in the post-processing instead.

- Preprocessing:

- o Specify the preprocessing techniques applied to the 10-meter SAR data. Given the resolution, speckle noise is a significant issue, and speckle filtering is essential. Clarify if and how this was handled.

Response: We use the data directly from the planetary computer (who gets data from Catalyst as a data processor), which uses standard SAR processing techniques such as orbit application, gamma correction, radiometric terrain correction using PlanetDEM, and speckle filtering. We've added a paragraph on this to the text in the 'Neural Network Model' subsection.

- o Did the study use ascending and descending passes, or only one of them? This should be stated explicitly.

Response: We utilize both ascending and descending passes to maximize temporal coverage, while ensuring we only compare image pairs with the same viewing geometry. We've added this clarification to the text.

- Feature Importance:

- o Consider performing a feature importance analysis to identify which auxiliary data contributes most to model performance. This would help reduce redundancy and improve the model's efficiency. Refer to studies that demonstrate the importance of feature selection in deep learning.

- o Specifically, it is necessary to determine the most important SAR features for flood mapping. For example, features like backscatter intensity, polarization (VV/VH), or temporal change metrics may vary in their significance depending on the region and type of flooding.

- o This approach could also guide the integration of auxiliary data, ensuring only the most impactful variables are included in the model. Refer to existing studies that highlight the benefits of feature selection in deep learning models.

Response: We want to clarify that our model uses only 4 input features from the SAR data:

- Binary indicator for whether the VV band is within the range of estimated water backscatter
- Binary indicator for whether the VH band is within the range of estimated water backscatter
- Delta in VV band intensity between the pre- and post-event imagery
- Delta in VH band intensity between the pre- and post-event imagery

We did experiment with including auxiliary data directly into the model, such as elevation, slope, and soil moisture, but did not see an improvement in the validation metrics. This is likely due to the lack of diverse flooding outcomes by slope (i.e., it is difficult to identify flooded areas in areas with high slope) and for soil moisture, the spatial resolution is so low that we essentially get one measurement per scene – it's not enough data for the model to learn from (this is already discussed in the Discussion). Given the lack of improvement by adding additional features, we stuck with the 4 dimensional input features.

Given the relatively low-dimensional input space, traditional feature importance analysis typically used for high-dimensional problems would add unnecessary complexity without providing meaningful insights. We acknowledge that some of the ideas presented could be included as additional features, but given the model's relatively strong performance (see updates on model validation), we leave these for future work.

In the new text, we have revised language on the features used and briefly described the tests on adding additional features and our justification for not including them in the final model.

- Reliability of Backscatter:

- o Acknowledge the limitations of SAR data on rainy days. Rain droplets and dew can alter backscatter values and impact classification accuracy. This should be explicitly discussed.

Response: As noted in our earlier response, we have modified the text in the introduction to acknowledge that SAR signals can be affected during extremely heavy rainfall. Specifically,

where we discuss SAR capabilities, we now state: 'SAR can penetrate through clouds and operate in both day and night conditions, though its signal may be affected during extremely heavy rainfall.' We believe this placement in the introduction, where we discuss the general capabilities and limitations of SAR technology, is appropriate for contextualizing these effects.

4. Differentiating Between Water Types:

- Provide an explanation of how the model differentiates:

- o Permanent surface water versus flooded areas.

- o Flooded cropland (e.g., rice paddies in Southeast Asia) versus actual flood events.

Response: The change detection approach naturally highlights only flooded areas vs permanent water. Qualitatively, permanent water will appear dark in both the pre- and post-flood SAR imagery, while flooded areas will only appear dark in the post-flood imagery. Additionally, we remove permanent water bodies during post-processing for any spurious flood detections that may occur. To better clarify this in the text, we've added a this to the Methods section on post-processing: "We also used the ESA land cover mapping to remove permanent water bodies like lakes and rivers. While our change detection approach should inherently ignore permanent water bodies since they appear as water in both pre- and post-event imagery, we found that explicitly masking them helped reduce noise in our predictions."

As for rice paddies, this is a great point. Our method does not differentiate between flooding caused by natural processes and manmade flooding. This is a limitation of our model and frankly most satellite detection models. We have added this important caveat to the Discussion section, noting that applications in regions with significant managed flooding (such as rice cultivation) would require additional contextual data or temporal analysis to distinguish between natural floods and agricultural practices. Here's the text we've added to the Discussion: "In particular, certain land use patterns require careful consideration when interpreting model results. A prime example is rice cultivation, where paddies are deliberately flooded as part of the agricultural cycle. While our model correctly identifies these areas as flooded, these detections represent intentional agricultural practices rather than natural flood events. Users should exercise particular caution when applying this model in regions with extensive rice cultivation or similar agricultural practices, where distinguishing between intentional and unintended flooding is crucial for proper interpretation."

- Special test cases should target regions prone to misclassification, such as paddy fields or other seasonally flooded croplands. For example, line 105 mentions that 11% of cropland lies in historically flooded areas—these regions need focused testing.

Response: As noted below, we have completely redone the model validation to include areas with paddy fields. The updated validation results can be found in the Model Validation section of the paper (with additional information in the Supplementary Information). Our response to this point can be found below under your comment 6. Testing and Evaluation.

The 11% figure specifically refers to cropland in flood-prone areas within our Ethiopian study region, based on discussions with local partners who have on-the-ground knowledge of agricultural practices in these areas. We have clarified this contextual detail in the text to avoid

any confusion about the scope of this statistic. We have also noted the crop types in this area and the fact that they are rainfed. As noted above, we have added a note to the Discussion about paddy fields and other deliberately flooded croplands.

5. Comparison and Validation:

- It would strengthen the study to compare the performance of:
 - o SAR-only models.
 - o Optical-only models.
 - o Fusion of SAR and optical data. This would provide insights into the complementary nature of these datasets for flood mapping.
- Discuss SAR's unique ability to detect dynamic water changes and compare it with prior studies.

Response: We agree that integrating SAR and optical data can potentially enhance flood mapping accuracy, and we acknowledge the benefits of such fusion approaches. While performing that comparison on our own work is beyond the scope of the current paper (see comment above), we have made additions to the text in the Discussion reflecting the relative strengths and weaknesses of the approaches, highlighting the complimentary nature of SAR and optical and the benefits of fusion models. And as noted above, because of the strengths of optical imagery (higher spatial resolution, true color representation plus infrared bands that can result in better detection of flooding), we are considering this for future work, repeating this type of analysis with both SAR and optical data.

6. Testing and Evaluation:

- The study needs to expand its testing regions, particularly targeting areas with complex hydrological dynamics (e.g., Southeast Asia, which has flooded cropland like rice fields). The current test cases are insufficient to generalize the findings to a global scale.

Response: Based on comments from multiple reviewers, we have done additional model validation. Based on the currently available datasets, we found that the recently published Kuro Siwo dataset to be the best match for our setup, since the dataset includes pre and post SAR imagery and expert-annotated labels. We've re-written much of the model validation section of the paper to reflect this new work. In summary, we find that in general our model performs well on this dataset, showing strong average performance across continents and similar aggregate performance to what the authors report in the Kuro Siwo paper, along with better performance than the Copernicus GFM ensemble.

We note that the Kuro Siwo dataset includes multiple scenes in south and southeast Asia that include rice fields. Performance in these scenes is quite good, with F1 scores ≥ 0.77 . While there is always more validation that can be done, we believe the performance of the model is the best that can be done with the currently available flood datasets.

Reviewer #3 (Remarks to the Author):

This paper describes the use of a ten-year record of Sentinel-1 SAR images to map floods globally, using a neural network along auxiliary dataset, including soil moisture, DEM, temperature, and land cover to minimize false positive errors.

It demonstrates trends at the global scale and assess some use cases around the world where the authors also validate the maps with high-resolution optical images from Planet Labs.

The paper is generally very well written and the methodology is sound, both technically and statistically.

Response: We thank the reviewer for their kind feedback!

In the main text of the paper, an in-depth comparison with and discussion of the Copernicus GFM S1 based flood maps, especially on the use cases presented would be much appreciated by a larger audience, I think. In my opinion, the authors should add this and also describe better the benefits and advantages/limitations of their global method compared to the Copernicus GFM.

In the Supplementary material. IOU (F1) is indeed low, maybe the authors should have some cases with S2 as a validation set and discuss the differences and thus maybe prove their point about the difficulty with high res optical images as a validation set.

Response: Based on feedback from all 3 reviewers, we have done extensive revisions to our Model Validation. Based on the currently available datasets, we found that the recently published Kuro Siwo dataset to be the best match for our setup, since the dataset includes pre and post SAR imagery and labels. We've re-written much of the model validation section of the paper to reflect this new work. In summary, we find that in general our model performs well on this dataset, showing strong average performance across continents and similar aggregate performance to what the authors report in the Kuro Siwo paper.

We also compare against GFM predictions and find that our model on average performs better than GFM. We do a detailed comparison for specific scenes in the Supplemental information, highlighting where our model outperforms GFM and where it doesn't, with implications for potential future work.

The previous model validation is still included in the Supplemental Information, but are not a key result in our work anymore.

Reviewer #3 (Remarks on code availability):

I saw no code being shared. My apologies if I missed this.

Response: We have a public github repository for the model here: <https://github.com/microsoft/ai4g-flood>

The model predictions are also freely available here: <https://huggingface.co/datasets/ai-for-good-lab/ai4g-flood-dataset>

Reviewer #1 (Remarks to the Author):

Well done to the authors for the pain-staking response to all the reviewers comments. I particularly appreciate the additional validation and inclusion of a exclusion mask.

The improvement to the Figures are appreciated. Personally I would like to see a scale bar on some of the more detailed map, but that's just being cartographically picky!

I find the large increase in flood detection from your method very intriguing!

It would be very interesting to evaluate your approach with this new Bayesian based work from Roth et al (2025) but that is probably out of the scope of this paper

Roth, F., Tupas, M.E., Navacchi, C., Zhao, J., Wagner, W. and Bauer-Marschallinger, B., 2025. Evaluating the robustness of Bayesian flood mapping with Sentinel-1 data: A multi-event validation study. Science of Remote Sensing, 11, p.100210.

We thank the reviewer for their kind words and thoughtful feedback. We agree that the Roth et al. (2025) paper presents a valuable example of detailed algorithm validation. While its specific focus lies beyond the scope of our global-scale approach, we have now cited the study and added a brief discussion of how similar evaluations could inform future work.

We also updated several figures to include scale bars and enhance visual clarity, a point raised by another reviewer.

We again thank the reviewer for their constructive comments and believe the paper is significantly stronger as a result.

Reviewer #1 (Remarks on code availability):

The code is hosted on github and is adequately documented

Reviewer #2 (Remarks to the Author):

Dear Authors,

I appreciate your effort in compiling a global flood extent dataset and acknowledge the importance of large-scale flood mapping initiatives. However, I would like to offer several comments and suggestions that I believe can help improve the clarity, rigor, and overall impact of your manuscript.

Response: We thank the reviewers for their helpful comments. We have carefully reviewed the comments below and adjusted the manuscript accordingly.

1. Limited Case-Based Evaluation One key concern is the generalization of algorithm performance based on a limited number of flood cases. It is difficult to conclude whether one algorithm is superior to

another without extensive and systematic validation across diverse flood events and environmental conditions. Many similar studies fall short in offering insights into the strengths and limitations of individual algorithms, making it challenging for readers to understand their applicability across different geographies and weather conditions.

I would strongly encourage the authors—and our research community at large—to place more emphasis on rigorous validation and physical understanding, rather than solely on developing increasingly complex algorithms. In this regard, I would like to highlight the exemplary work by Florian Roth et al. (2025), who conducted a robust multi-event evaluation of a Bayesian flood mapping algorithm using 18 flood events across five continents with Sentinel-1 data. His study offers critical insights into the sensitivity of VV and VH polarizations over vegetated flood areas and identifies important limitations in existing approaches. This paper is currently missing from your references and I highly recommend reviewing it:

Roth, F., Tupas, M.E., Navacchi, C., Zhao, J., Wagner, W., Bauer-Marschallinger, B., 2025. Evaluating the robustness of Bayesian flood mapping with Sentinel-1 data: A multi-event validation study. *Science of Remote Sensing*, 100210. Link

Notably, this work has already informed improvements in the operational Copernicus Global Flood Monitoring (GFM) service.

Response: We thank the reviewer for their constructive comments and for highlighting the recently published Roth et al. (2025) paper. We agree this is an important study demonstrating a valuable approach for in-depth algorithm evaluation, providing critical insights into specific sensitivities like VV vs VH polarization. We have now cited Roth et al. in the manuscript and incorporated it into our discussion, including acknowledging that such detailed investigation represents a potential avenue for future work building upon our findings.

Regarding the validation of our approach, we wish to respectfully clarify that our validation, significantly expanded in the previous revision based on valuable reviewer feedback (detailed in Methods section (subsection ‘Model Validation’) and SI Section S1.5.2), is already quite extensive. It utilizes the comprehensive Kuro Siwo benchmark dataset, which encompasses 43 diverse flood events spanning 6 continents. We selected this broad benchmark specifically because our primary contributions are the introduction of a novel model intended for global-scale application and the creation of a decade-long global dataset. While the Roth et al. (2025) study provides crucial insights via a deep dive into a specific algorithm's performance sensitivities across 18 events – a valuable approach we now cite and discuss as a potential direction for future, focused investigations (see Discussion, page 13, lines 353-362) – we believe our validation across 43 diverse global events robustly assesses our model's performance and generalizability for the broad geographic scope and stated objectives of this manuscript.

2. Motivation for Temporal Aggregation While aggregating 10 years of flood extent data may be useful from a historical perspective, the added value is not fully clear in your current discussion. What does this aggregated product offer, and how can it be linked to broader trends such as climate change, land use changes, or other hydrometeorological drivers? Without such context, it is difficult to interpret the observed increase or decrease in flood extents across regions.

Response: We thank the reviewer for highlighting the need to more clearly articulate the motivation and added value of our 10-year temporal aggregation. We agree this is crucial context and understand the concern relates to clarifying the specific contributions of both the aggregated historical map and the longitudinal trend analysis derived from it.

To address this, we have made specific revisions to the manuscript to better articulate the unique value proposition of this decade-long, SAR-based dataset:

1. **Revised Introduction:** We have significantly revised the paragraph outlining the motivations and use cases enabled by our approach (see revised text on page 2, lines 78-92). This revision now more explicitly states the value of generating:
 - A comprehensive, high-resolution historical baseline map using cloud-penetrating SAR, emphasizing its importance for robust risk assessment and its complementarity to optical datasets often affected by cloud cover during floods.
 - An observation-based trend analysis, highlighting its contribution as the first globally consistent SAR perspective on potential decadal dynamics, establishing a crucial baseline for future monitoring, and offering a less biased view compared to report-based methods, while carefully maintaining appropriate caution regarding climate attribution given the 10-year record length.
2. **Enhanced Discussion:** We have also reinforced these points within the Discussion section:
 - In the paragraph discussing the benefits of our approach over existing datasets (page 12, lines 312-322), we have added a sentence explicitly connecting SAR's cloud-penetrating capability to the value of creating the robust historical baseline map mentioned in the Introduction.
 - We have rebalanced the paragraph focused on the trend analysis limitations (page 14, lines 388-398). This section now leads by reiterating the positive contribution and value of the trend analysis (e.g., first SAR-based observation look at flooding over time, baseline for future monitoring) before detailing the necessary and important caveats regarding the limitations for climate attribution over the current time frame. The concluding sentences have also been strengthened to emphasize the foundation this work lays for future long-term monitoring.

3. Specific Comments on the Manuscript

- Lines 33–36: This sentence needs rephrasing for clarity and to avoid redundancy.

Response: Thank you for pointing this out. We have revised the sentences describing optical/infrared and SAR sensors (original Lines 33-36) for better clarity and flow, removing the potentially redundant sentence. The revised text can be found on page 1, lines 32-42.

- Lines 39–40: The claim that "daily observations from constellations" are available is incorrect. For HLS (Harmonized Landsat-Sentinel), the typical revisit time is 2–3 days, not daily.

Response: We thank the reviewer for pointing out the potential ambiguity in our original statement about 'daily coverage' and for providing the example of the HLS constellation's 2-3 day revisit time. We agree that clarification was needed for greater accuracy. Our initial statement was based on the revisit time of VIIRS and commercial satellite offerings such as Planet labs, which do have daily optical data.

To address this, we have revised the sentence in the Introduction (page 1, line 39) to more accurately reflect the *range* of optical revisit frequencies. The revised text now clarifies that revisit times typically range from near-daily to several days, providing better context for the comparison with Sentinel-1 SAR.

- Lines 66–67: Please clarify the specific benefit of the proposed approach in this context.
- Line 72: What is the practical advantage of creating a 10-year SAR-based flood extent database, especially considering Sentinel-1's limitations (e.g., revisit time, difficulty with forested flood areas)?
Response: We thank the reviewer for these two related points requesting clarification on the specific benefit and practical advantage of our 10-year aggregated SAR dataset approach (around Lines 66–72 in the original), particularly in light of Sentinel-1's known limitations. These comments are closely tied to the overall motivation for our work, which we also address in Point 2.

As detailed in our response to Point 2, we have substantially revised the Introduction paragraph (page 2, lines 78-92) that outlines the motivations and use cases for our approach. This revised section, which follows the context around Lines 66–72 in the original (now lines 62-70), clearly articulates the specific benefits and practical contributions of our work—including the creation of a unique, cloud-resilient historical baseline and the ability to perform observation-based trend analysis. We believe this added explanation directly addresses the core questions raised in both reviewer comments 3.3 and 3.4.

Regarding the specific Sentinel-1 limitations mentioned (e.g., revisit frequency, challenges in forested/complex terrain), we agree these are important considerations and are acknowledged in the manuscript. The implications of revisit time are discussed in the Introduction (page 1, lines 38-42) when comparing sensor types. Challenges related to complex terrain (including forests), along with the rationale and implications of our exclusion mask and its impact on validation scope, are addressed in the Methods (page 17, lines 550-553) and Discussion (page 13, lines 341-345).

- Line 83–84: While historical flooding can indicate areas of potential future risk, it is important to acknowledge the role of climate change and other dynamic factors that may alter future flood patterns. This aspect is missing from your discussion.

Response: We thank the reviewer for highlighting the importance of explicitly acknowledging that future flood patterns may be altered by dynamic factors such as climate change. We agree that this is an important caveat when discussing the utility of historical flood data.

To address this, we have revised the relevant sentence in the Introduction's motivation paragraph (page 2, lines 82-85) to explicitly incorporate this point. The updated sentence now reads:

“While historical flooding is not a perfect predictor of future risk, particularly as climate change and other dynamic factors (e.g., land use change) may alter future patterns, this baseline can inform risk assessments, mitigation planning, and resilient infrastructure development.”

4. Issues with the Introduction and Figures

- The introduction could be improved for clarity and conciseness. The last paragraph, in particular, repeats earlier points and would benefit from restructuring.

Response: We agree with the reviewer on the need for improvement in the last paragraph. As detailed in our responses to Point 2 and Point 3 above, the final paragraph of the Introduction outlining the motivations and practical advantages of our approach has been substantially revised and restructured (page 2, lines 78-92).

- Figure 1: The color scheme and visual quality are suboptimal. The figure is difficult to interpret, and the visualization should be improved.

Response: We appreciate the reviewer's feedback regarding the clarity of Figure 1. We agree that the exclusion mask's light gray tone in the original figure may have been difficult to distinguish from other background elements at a global scale. We experimented with alternative color schemes, including tan and blue hues, but found they competed visually with the flood detections or distracted from the main focus of the map. As a result, we revised Figure 1 to use a darker gray for the exclusion mask, improving contrast while preserving visual balance and interpretability. We have also updated the other figures (2, 3, and 5) that have an exclusion mask layer.

- The masking of regions (due to urbanization, desert areas, etc.) appears to have excluded countries like Iran, which have experienced significant flood events. Please explain the rationale and implications of this exclusion.

Response: We thank the reviewer for raising this concern. We agree that at global scale, the exclusion mask can give the impression that entire countries—such as Iran—are fully excluded from the analysis.

However, our exclusion mask operates at the pixel level, based on land surface properties like terrain steepness, aridity, and urbanization. While parts of Iran, particularly its central desert and some mountainous regions, are masked due to SAR limitations in those environments, flood detection is still performed in all unmasked areas within the country.

To clarify this point, we include below a zoomed-in map of Iran showing flood detections (blue) and excluded areas (gray). As this plot illustrates, substantial regions—especially in the west and along rivers—are *not* excluded and show active flood detections in our dataset.

We hope this visualization clarifies that countries like Iran are not entirely excluded, and that our masking approach preserves analysis in regions where SAR is effective. Given the pixel-level basis of the exclusion mask, this pattern is common in geographically diverse countries.

- The validation approach remains unclear, particularly for flood-prone areas such as mountainous regions and flooded forests, which are often excluded or poorly captured.

Response: We thank the reviewer for raising this important point. As noted in the Methods section and further elaborated in the Discussion, we explicitly mask out areas where SAR flood detection is known to be unreliable due to terrain or land cover complexity — including steep mountains, dense forests, and urban areas — using an exclusion mask (see Post-Processing subsection). These areas are not only excluded from our flood maps but are also excluded from our validation exercises.

To address the reviewer’s concern more directly in the text, we have revised the paragraph in the Model Validation section (page 13, lines 550-553) to clarify this point. Specifically, we now note that while the Kuro Siwo dataset includes diverse global flood events, the validation focuses on regions where our model is expected to perform reliably. The new sentence reads:

“Given the known challenges of SAR in complex terrains like steep mountains and dense forests — areas where our model’s predictions are masked using the exclusion mask (see Methods subsection on Post-

Processing) — this validation focuses on assessing model performance in the diverse global regions where detections are expected to be most reliable.”

- Lines 153–163: The justification provided for why your approach captures "additional" flood areas more accurately than previous methods is not convincing. More evidence or validation is needed.

Response: We thank the reviewer for pushing for a more convincing justification regarding the accuracy of the additional flood areas captured by our method (around Lines 153-163 in the original). We agree that citing SAR's cloud-penetration capability alone is insufficient and that providing stronger, multi-faceted evidence is necessary.

Our confidence in the accuracy of these additional detections stems from several converging lines of evidence, which we have now made more explicit in the revised Results section (page 4, lines 149-164 in the revised version):

1. Mechanism: We reiterate that SAR's ability to penetrate cloud cover provides the mechanism by which it can detect events missed by optical sensors.
2. Quantitative Validation: We now explicitly reference the model's strong performance in quantitative validation, particularly its high recall demonstrated on the diverse Kuro Siwo benchmark dataset (72%), indicating its general effectiveness in identifying true flood pixels when present.
3. Qualitative Local Validation: Crucially for the Ethiopia examples discussed in this section, our assessment was corroborated through qualitative validation conducted with local partner organizations (IOM Ethiopia), whose deep domain knowledge confirmed the plausibility and accuracy of the additional flood patterns identified by our model in these specific regions. This collaborative validation process and associated findings are documented in a related technical report (<https://environmentalmigration.iom.int/sites/g/files/tmzbd1411/files/documents/2024-11/vulnerable-communities-ethiopia-communities-at-risk-of-flooding-final.pdf>), which we now cite in the manuscript (page 4, lines 144 and 157).

Together, these combined lines of evidence—the physical mechanism (cloud penetration), quantitative performance metrics, and qualitative validation by local experts—provides the robust and convincing justification requested by the reviewer.

- Figure 8: To enhance its value, consider linking it to climate-related drivers of flood change. Without this context, the figure’s contribution to advancing flood mapping knowledge remains limited.

Response: We thank the reviewer for the suggestion to enhance the value of Figure 8 by considering links to potential drivers of flood change. We agree that providing context is valuable for interpretation.

As noted in the manuscript text, we maintain scientific caution regarding definitive attribution of the observed 10-year trends, given the challenges of distinguishing long-term signals from climate variability and episodic events over this relatively short timeframe.

However, to explicitly address the reviewer's suggestion and better frame the figure's contribution toward future understanding, we have revised the caption for Figure 8 (page 11, lines 292-299). The new concluding sentence reads: "These regional patterns merit follow-up in future work, especially to assess whether any align with climate-related drivers as longer observational records become available."

Reviewer #3 (Remarks to the Author):

I have read the responses to comments and the revised version, and I thank the authors for their extensive revisions.

In my opinion, the responses and revisions with regard to my points of concern raised are adequate and sufficient.

We thank the reviewer for their kind words and for their previous comments which helped strengthen the paper!

Reviewer #3 (Remarks on code availability):

I have not reviewed the code. I do not feel in the best position to review computer code in the sense of bugs and inconsistencies.

I would like to thank the authors for their thoughtful responses to my previous comments. I appreciate the effort they have put into revising the manuscript and addressing the concerns raised.

Response: We thank the reviewer for their helpful comments throughout the process. We think the paper is substantially stronger because of their suggestions.

That said, for several of the comments, I had expected more substantial revisions—particularly beyond citing prior work or adding a few lines in the introduction or discussion. With all due respect to the authors' efforts and the use of an extensive 10-year Sentinel-1 dataset, I am unable to identify a clear added value or novel contribution that significantly advances the field or benefits future research. As such, I remain unconvinced of the broader impact of this study on the SAR flood mapping community.

Although I have no further technical comments at this stage, I do not believe the manuscript meets the threshold for publication in its current form. Given the authors' considerable efforts, I defer the final decision to the editor and would also welcome input from an additional reviewer for a broader perspective.

Response: Regarding the reviewer's comments on the broader impact and novelty: while we understand their perspective, we have aimed in the revised manuscript to clearly articulate the primary contributions. These include the generation of a globally consistent 10-year SAR-based flood dataset which identifies flood-prone areas often missed in existing datasets (as quantified in Table 1 and shown for specific regions in Figure 3), and the longitudinal analysis of global flood trends derived from this SAR data (Figures 6 and 7, Table 2). We also highlight the public release of our model predictions and codebase.

We note the reviewer's confirmation that all technical points are resolved, and we appreciate them deferring the final decision to the editor.